# Electrochemical oxo-functionalization of cyclic alkanes and alkenes using nitrate and oxygen

Joachim Nikl[1], Kamil Hofman[1], Samuel Mossazghi[1], Isabel C. Möller[1], Daniel Mondeshki[1], Frank Weinelt[2], Franz-Erich Baumann[2] & Siegfried R. Waldvogel [1] ✉

Direct functionalization of C(sp³)–H bonds allows rapid access to valuable products, starting from simple petrochemicals. However, the chemical transformation of non-activated methylene groups remains challenging for organic synthesis. Here, we report a general electrochemical method for the oxidation of C(sp³)–H and C(sp²)–H bonds, in which cyclic alkanes and (cyclic) olefins are converted into cycloaliphatic ketones as well as aliphatic (di)carboxylic acids. This resource-friendly method is based on nitrate salts in a dual role as anodic mediator and supporting electrolyte, which can be recovered and recycled. Reducing molecular oxygen as a cathodic counter reaction leads to efficient convergent use of both electrode reactions. By avoiding transition metals and chemical oxidizers, this protocol represents a sustainable oxo-functionalization method, leading to a valuable contribution for the sustainable conversion of petrochemical feedstocks into synthetically usable fine chemicals and commodities.

Transforming a chemically inert C(sp³)–H bond into a functional group is described as a vibrant topic, a great challenge, and even as a holy grail for synthetic organic chemistry[1–3]. Indeed, functionalization and modeling of molecular structures starting from hydrocarbons require the cleavage of carbon–hydrogen or carbon–carbon bonds. Due to the limited preparative application fields, these compounds are commonly used as fuels[4] and organic solvents[5]. Furthermore, alkanes' missing functionalities and lipophilic nature lead to ecological issues since their bioavailability for microorganisms is poor[6]. Naturally occurring monocyclic alkanes were discovered in the 1890s by Markovnikov in Caucasian crude oil called naphtha[7,8]. Cyclohexane, the most prominent example and ubiquitous structural motif in natural and synthetic compounds, is industrially produced via the hydrogenation of benzene[9]. Monocyclic aliphatics with larger ring sizes, especially eight and twelve-membered rings, are formed through cyclooligomerization of butadiene and subsequent hydrogenation[10]. While the fully saturated cyclododecane serves as an essential intermediate for the production of laurolactame via a Bashkirov oxidation to primarily cyclododecanol[11], other saturated cyclic hydrocarbons emerge as by-products and are usually incinerated or discarded. The Bashkirov oxidation mentioned above produces additional reagent waste due to the use of boric acid, which is toxic to reproduction and fertility[12], making it necessary to improve this process towards greater sustainability. Apart from that example, the unsaturated representatives of the same ring size are usually of substantial commercial interest. Cyclohexene, cyclooctene, and cyclododecene serve as important starting materials for synthesizing distinct polymer precursors like dicarboxylic acids, annually produced in several ten thousand tons with strongly increasing demand[13,14]. The incorporation of saturated hydrocarbons back into the value chain provides both economic and also environmental advantages. Therefore, valorization by installing an oxygen-containing group like a carbonyl function leads to synthetic versatility[15].

Over the past decades, different oxo-functionalization methods have been developed (Fig. 1a). Conventional methods for the oxidation of cycloalkanes are predominantly based on transition metal-catalyzed

[1]Department of Chemistry, Johannes Gutenberg University Mainz, Duesbergweg 10–14, 55128 Mainz, Germany. [2]Evonik Operations GmbH, Paul-Baumann-Strasse 1, 45772 Marl, Germany. ✉e-mail: waldvogel@uni-mainz.de

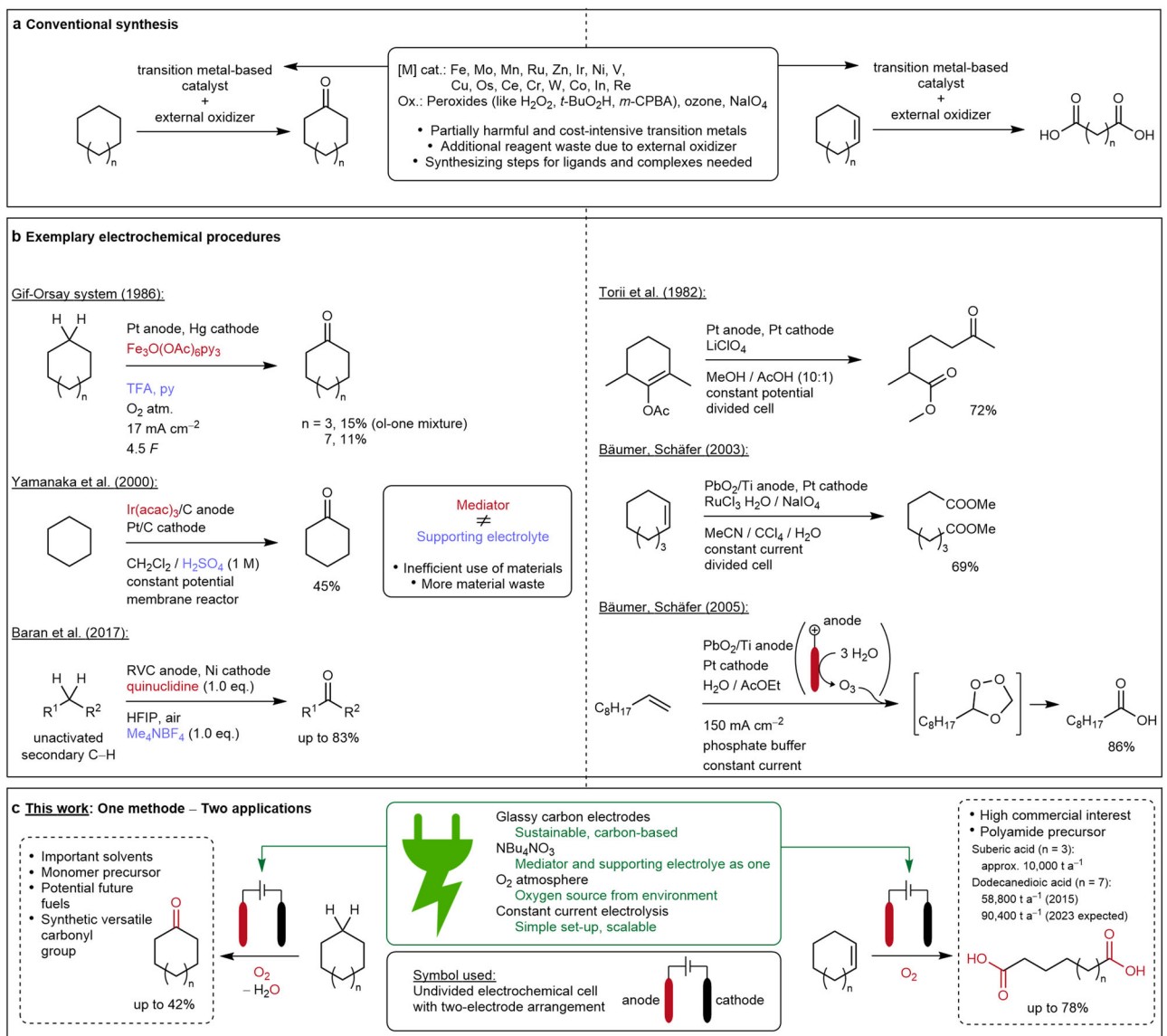

**Fig. 1 | Overview of known procedures and the subject of this work.**
**a** Conventional synthesis protocols for converting cycloaliphatics to ketones and dicarboxylic acids. **b** Known electrochemical procedures for oxo-functionalizing C(sp³)−H and C=C bonds to ketones and dicarboxylic acids. Mediators labeled in red and supporting electrolytes are labeled in blue.
**c** Subject of this work: application of one electrochemical method to valorize saturated and unsaturated cycloaliphatics. Oxygen as $O_2$ source and oxo-groups are labeled in red.

reactions in combination with chemical oxidizers, mainly peroxides. Frequently, hydrogen peroxide is reported with various catalytically active transition metals such as iron[16], manganese[17], copper, nickel, and vanadium[18]. Ruthenium often serves as a transition metal catalyst in the presence of peracids[19] or *tert*-butyl hydroperoxide[20]. Popular ligand systems are based on porphyrins, including manganese or iron centers, with peroxide species[21] and oxygen[22] as oxidizers. Examples of an iridium catalyst with sodium periodate[23] and an iron(III) nitrate/*N*-hydroxyphthalimide/oxygen system[24] have also been reported.

With regards to C=C double bond cleavage to the corresponding dicarboxylic acids, many conventional methods are already known. Comparable to those previously mentioned, these usually occur in a transition metal-catalyzed manner. Here, osmium[25,26], ruthenium[27,28], and tungsten[29,30] catalysts are most frequently described. Older protocols refer to the use of potassium permanganate[31]. In contrast to these, metal-free processes are also known. For example, via hyper-valent iodine species in combination with peracids or oxone[32,33], or classically, via the ozonolysis reaction[34].

Electrochemical protocols for oxidizing C(sp³)−H bonds to ketones have scarcely been described (Fig. 1b). The Gif-Orsay system, developed in the 1980s, serves as an example[35,36]. Also, using an elec-trocatalytic $MnO_x$ layer on a titanium anode with oxygen[37] or the oxi-dation of cyclohexane with water on an iridium-supported carbon anode[38] was reported. The group of Baran published 2017 an electro-chemical protocol for C(sp³)−H oxidation via quinuclidine as a med-iator in the presence of oxygen[39]. Adversely, the method is unsuitable for technical scale applications as the solvent, 1,1,1,3,3,3-hexa-fluoropropan-2-ol (HFIP), has approximately 200 times higher global warming potential ($GWP_{100}$) than carbon dioxide[40]. Just as with alkanes, only a few electrochemical methods for the double bond cleavage to dicarboxylic acids are known (Fig. 1b), as shown by Schäfer and Bäumer via ruthenium catalysis with electrochemically regener-ated periodate[41]. The authors also described electrochemical ozono-lysis from water using a lead dioxide anode[42]. Older electrochemical methods use methanol as an oxygen source at a constant potential in a divided cell[43]. The disadvantages of the methods mentioned above

relate to using toxic or, due to their mining, socially and environmentally problematic transition metals[44]. Furthermore, the use of ligands and greater than a stoichiometric amount of oxidizers leads to reagent waste. The known electrochemical processes often use expensive or toxic electrode materials or require a high material input due to the separate roles of electrochemical mediator and supporting electrolyte, devaluing the economic efficiency of these protocols.

With the method presented in this work, both a direct oxo-functionalization of chemically stabile $C(sp^3)–H$ bonds and a cleavage of C=C double bonds to carbonyl and carboxyl compounds can be implemented by using electric current as a clean and safe reagent[45] (Fig. 1c). In terms of sustainability and resource preservation, the use of transition metals and additional oxidizing agents is avoided[46,47]. Furthermore, the method is characterized by the effective use of materials through the dual use of the supporting electrolyte as an electrochemical mediator[48]. Here, the initial anodic oxidation of nitrate anions serves as a radical source, inspired by literature reports[49–51]. Commonly described oxygen reduction to superoxide radicals was found to be the cathodic counter reaction[52,53], completing the electric cycle in a convergent-type electrolysis[54]. Advantageously, the supporting electrolyte can be recovered by extraction and reused for electrolysis, which further supports the sustainability of this electrochemical procedure (see Supplementary Results 2.7). The products are important substrates for organic synthesis and monomer building blocks for polymers and, therefore, highly relevant for industrial applications.

## Results

### Reaction development

Electrochemical oxidation of saturated hydrocarbons is a very challenging task for several reasons. Direct oxidation of these chemically inert species on the electrode surface is difficult due to their high oxidation potentials[39]. Usually, the electrolyte, consisting of the solvent and the supporting electrolyte salt, is oxidized prior to these substrates. Furthermore, polar solvents with high permittivity are required to avoid high ohmic resistance within the electrochemical cell. Very lipophilic substrates like saturated hydrocarbons frequently display poor solubility in these solvents. Therefore, a suitable strategy is a mediated electrochemical system in which the mediator is transformed into a highly reactive species that allows the target reaction to occur.

As a set-up, a commercially available electrochemical screening system from IKA was used, which was co-developed[55] (see Supplementary Fig. 1). Inspired by literature[51], we focused upon the use of nitrate salts as supporting electrolyte and electrochemical mediator in a dual role since upon oxidation the highly reactive nitrate radicals can split $C(sp^3)–H$ bonds. In initial reactions of converting cyclooctane (**1c**) as a model substrate to cyclooctanone (**2c**), tetra-butylammonium (TBA) nitrate proved itself suitable. Commercially available acetonitrile, a common solvent in electro-organic applications, was used without further purification. Glassy carbon was used as a robust, long-term durable electrode material with excellent electric conductivity properties.

The first experiments were conducted with ambient air in the gas space above the reaction solution. Due to the manufacturing design, the lid used for the electrochemical cell ensures an air exchange with the environment. Cyclooctane (**1c**) was electrolyzed using 10 mA cm$^{-2}$ and a charge of 4 $F$, leading to a 23% yield of **2c** (Table 1, Entry 2). The reaction proved selective, as only cyclooctanol (**3c**) and cyclooctane-1,4-dione (**4c**) were obtained as significant by-products with 2% and 3% yields, respectively. If the reaction was carried out at 100 vol% of $N_2$, no reaction to oxidized species occurred (Table 1, Entry 3), which underlines the necessity of $O_2$ in the reaction medium. Increasing the $O_2$ content in the atmosphere to 100 vol% remarkably led to only 16% of **2c** (Table 1, Entry 1). The decreased yield is assumed to be caused by mutual influences of the atmospheric $O_2$ amount and the applied

current density. However, a maximum yield of 31% was obtained for **2c** under 20 vol% $O_2$ and 10 mA cm$^{-2}$ (Table 1, Entry 4).

The necessity of nitrate as the anion is demonstrated by comparison with other TBA salts. Tetrafluoroborate ($BF_4^-$), hexafluorophosphate ($PF_6^-$), and perchlorate ($ClO_4^-$) anions only lead to a formation of **2c** in yields of 3–4%. Apart from TBA as the cation, also longer chained tetra-alkylammonium nitrate salts provided product formation in comparable yields of 17–28% (see Supplementary Table 1), and also tetra-butylphosphonium nitrate was suitable with 20% (Table 1, Entry 7). Furthermore, the reaction takes place in different solvents like isobutyronitrile (*i*-PrCN) (24%), acetone (29%), and nitropropane (17%) (see Supplementary Table 1). The stirring rate significantly influences the reaction as a yield drop appears at higher and lower stirring rates than 350 rpm (see Supplementary Table 1). This circumstance is attributed to a disturbed $O_2$ adsorption on the electrode at higher stirring rates and the reduced reaction partner contact due to lower stirring rates. The reaction also occurs with lower yields at different carbon-based electrodes like boron-doped diamond (BDD) and graphite (see Supplementary Table 1).

Application of the same methodology to cyclic alkenes **5** revealed that they undergo C=C double bond cleavage. Following the observation of a dicarboxylic acid **6** via HPLC-MS, the reaction conditions were varied using cyclooctene (**5b**) as a model substrate. Applying the same standard conditions as for the cyclooctane oxidation led to a 33% yield of suberic acid (**6b**), while lowering the temperature to 5 °C led to 47% (Table 1, Entries 1 and 8). Increasing the charge to 8 $F$ caused a yield drop to 28% (Table 1, Entry 9). The exclusion of $O_2$ showed no formation of a dicarboxylic acid, indicating the necessity for atmospheric oxygen, like for the ketone formation (Table 1, Entry 3). Decreasing the substrate concentration and the current density to 0.05 mol L$^{-1}$ and 5 mA cm$^{-2}$ by varying the applied charge from 4 $F$ to 6 $F$ and 8 $F$ led to similar yields of 39–44% (Table 1, Entries 10, 11, and 12). Using isobutyronitrile as a solvent did not influence the yield (Table 1, Entry 13).

Besides the reaction development trials on **5b**, a further intensified optimization has been carried out with cyclododecene (**5c**) since it results in the highly industry-relevant dodecanedioic acid (**6c**) (see Supplementary Table 2). Due to the poor solubility of **5c** in acetonitrile, the reactions were conducted in isobutyronitrile. Decreasing the substrate concentration and the current density to 0.05 mol L$^{-1}$ and 5 mA cm$^{-2}$ led to the best conditions for forming **6c** with a yield of 78%. As molecular oxygen can act as an oxidizing agent, it is noteworthy that the reaction does not occur if no electricity is applied to the reaction mixture (see Supplementary Table 2).

### Scope, batch-type, and flow electrolysis

After varying several reaction parameters with the model substrates for both reaction types, scale-up reactions, and different substrates were tested to investigate the method's applicability. A scale-up experiment regarding the cyclooctane oxidation was performed in a 50 mL round-bottom flask under an air atmosphere providing ketone **2c** in a yield of 35% (see Supplementary Fig. 2a). However, an improved yield of **2c** could be achieved by using a three-necked 100 mL round-bottom flask (see Supplementary Fig. 2b) and pure oxygen atmosphere, to receive **2c** in a yield of 42%. The increased yield can be explained by the different cell set-up allowing a larger contact area between the oxygen-containing atmosphere and the reaction solution. Next, we used this set-up to explore the oxidation of cyclic alkanes containing six to twelve methylene groups (Fig. 2a). In comparison, the yield of cyclooctanone (**2c**) stayed the highest. Smaller and larger ring sizes seem more challenging to convert as mostly starting material remained after electrolysis. Remarkably, compared to the increasing ring sizes, the general trend regarding the ketone yields is approximately following the transannular Prelog strain[56], as cyclohexanone (**2a**, 6% yield) and cyclododecanone (**2e**, 4% yield) gave the lowest

**Table 1 | Chosen condition examples for the reaction development**

Standard reaction conditions:

glassy carbon anode / glassy carbon cathode

- undivided 5 mL PTFE cell,
- acetonitrile (5 mL),
- substrate (0.2 mol L$^{-1}$),
- NBu$_4$NO$_3$ (0.5 eq.),
- 30–35 °C,
- O$_2$ atmosphere (100 vol.%),
- 350 rpm,
- 4 $F$ (ref. substrate),
- 10 mA cm$^{-2}$

| Entry | Deviations from conditions | 6b[a] | 2c[b] | 3c[c] | 4c[c] |
|---|---|---|---|---|---|
| 1 | None | 33% | 16% | 1% | 1% |
| 2 | Air | 40% | 23% | 2% | 3% |
| 3 | O$_2$/N$_2$ = 0/100 | 0% | 0% | 0% | 0% |
| 4 | O$_2$/N$_2$ = 20/80 | 37% | 31% | 1% | 6% |
| 5 | O$_2$/N$_2$ = 20/80, NBu$_4$PF$_6$ | Traces | 3% | 2% | 1% |
| 6 | O$_2$/N$_2$ = 20/80, acetone | Traces | 29% | 2% | 4% |
| 7 | PBu$_4$NO$_3$ | 21% | 20%[d] | 0% | 7% |
| 8 | 5 °C | 47% | 14%[d] | 1% | 1% |
| 9 | 5 °C, 8 $F$ | 28% | 18%[d] | 0% | 9% |
| 10 | substrate (0.05 mol L$^{-1}$), 5 mA cm$^{-1}$ | 41% | 12%[d] | 1% | 1% |
| 11 | 6 $F$, substrate (0.05 mol L$^{-1}$), 5 mA cm$^{-1}$ | 44% | 14%[d] | 1% | 1% |
| 12 | 8 $F$, substrate (0.05 mol L$^{-1}$), 5 mA cm$^{-1}$ | 39% | 15%[d] | 1% | 2% |
| 13 | $i$-PrCN, substrate (0.05 mol L$^{-1}$), 5 mA cm$^{-1}$ | 46% | 15%[d] | 1% | 1% |

Oxygen as O$_2$ source and oxo-groups are labeled in red. $i$-PrCN: isobutyronitrile. For preparative information, see Methods: "Electrolysis in 5 mL PTFE cells (GP 1)".
[a]Isolated yields.
[b]$^1$H NMR yield (internal standard: 1,3,5-trimethoxybenzene).
[c]GC yields are calculated based on the yield of **2c**.
[d]GC yields (external calibration of **2c**, 1,3,5-trimethoxybenzene as internal standard).

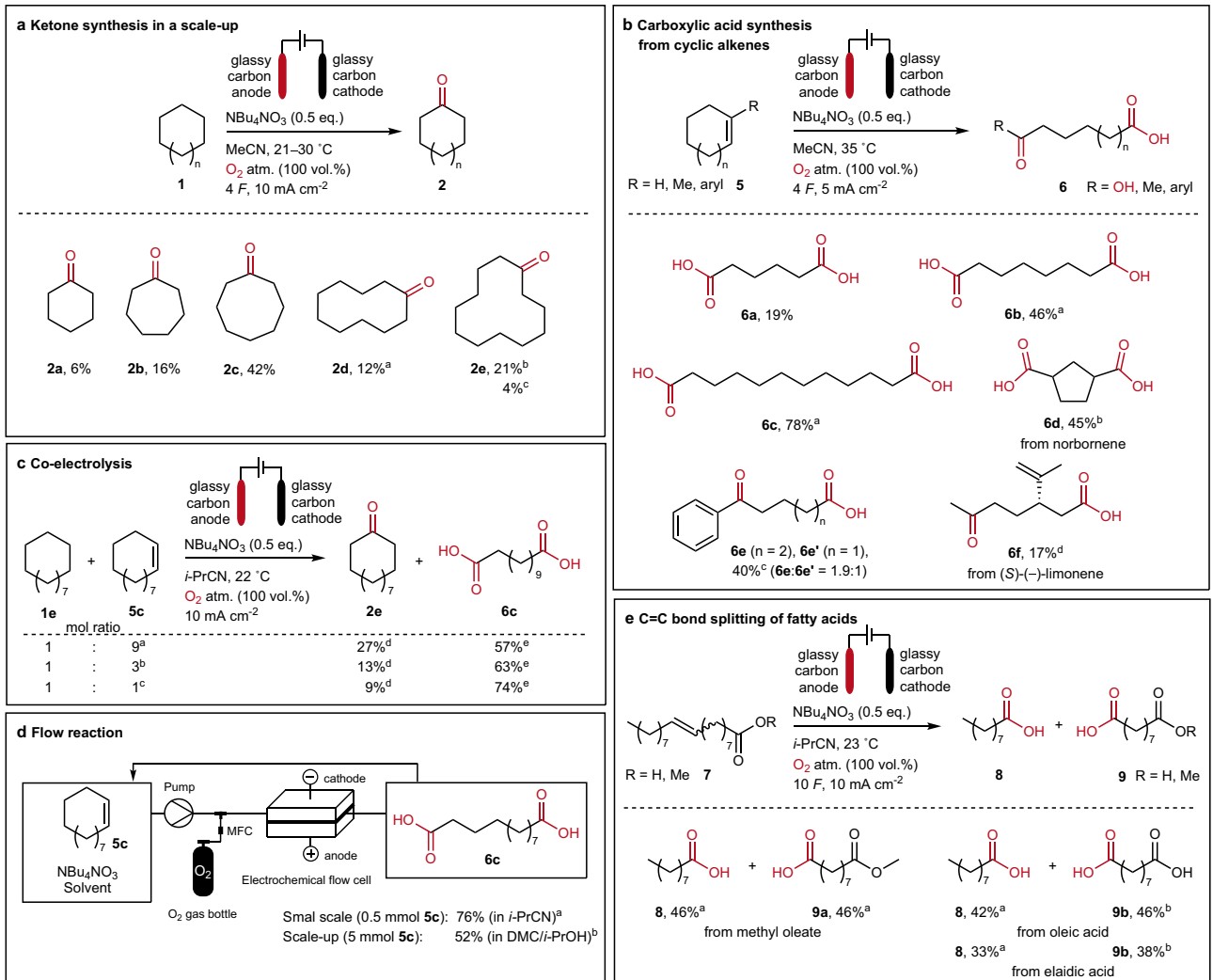

**Fig. 2 | Scope of ketones and (di)carboxylic acids, including batch and flow processes.** Unless separately indicated, all yields are isolated. Oxygen as $O_2$ source and oxo-groups are labeled in red. atm.: atmosphere. **a** Conditions: 100 mL round-bottom flask, acetonitrile (MeCN, 25 mL), substrate (0.2 mol $L^{-1}$), 400 rpm. [a]5 $F$. [b]Isobutyronitrile. [c]50 °C, air atmosphere. **b** Conditions: undivided 5 mL PFTE cell, acetonitrile (MeCN, 5 mL), substrate (0.05 mol $L^{-1}$), 350 rpm. [a]Isobutyronitrile. [b]5 °C. [c]Combined yield ratio determined via [1]H NMR. [d]25 °C. **c** Conditions: undivided 5 mL PFTE cell, isobutyronitrile ($i$-PrCN, 5 mL), NBu$_4$NO$_3$ (0.1 mol $L^{-1}$, 0.5 eq. toward combined mol of **1e** and **5c**), 350 rpm. [a]**1e** 0.02 mol $L^{-1}$, **5c** 0.18 mol $L^{-1}$, 7,6 $F$. [b]**1e** 0.05 mol $L^{-1}$, **5c** 0.15 mol $L^{-1}$, 7 $F$. [c]**1e** 0.1 mol $L^{-1}$, **5c** 0.1 mol $L^{-1}$, 6 $F$. [d]Yields are determined via GC calibration and refer to mol% of **1e**. [e]Isolated yields refer to mol% of **5c**. **d** Flow reactions were carried out using an IKA flow cell with a 2 × 6 cm² anode surface. [a]Conditions: glassy carbon anode and cathode, isobutyronitrile ($i$-PrCN, 10 mL), substrate (0.05 mol $L^{-1}$), NBu$_4$NO$_3$ (0.5 eq.), 20 °C, $O_2$ atmosphere (100 vol%), electrolyte flow: 10 mL min$^{-1}$, gas flow: 10 mL min$^{-1}$, 4 $F$ (ref. substrate), 5 mA cm$^{-2}$. [b]Conditions: glassy carbon anode and cathode, dimethyl carbonate (DMC)/isopropanol ($i$-PrOH) 9:1 (9.12 mL), substrate (0.5 mol $L^{-1}$), NBu$_4$NO$_3$ (1 eq.), 50 °C, $O_2$ atmosphere (100 vol%), electrolyte flow: 18 mL min$^{-1}$, gas flow: 20 mL min$^{-1}$, 2 $F$ (ref. substrate), 20 mA cm$^{-2}$. MFC: mass flow controller. **e** Conditions: undivided 5 mL PFTE cell, isobutyronitrile ($i$-PrCN, 5 mL), substrate (0.1 mol $L^{-1}$), 350 rpm. [a]Yields determined via GC calibration. [b]Isolated yields.

yields compared to cyclooctanone (**2c**, 42% yield). Because of the poor solubility of cyclododecane (**1e**) in acetonitrile, the solvent was replaced by isobutyronitrile, yielding 21% of cyclododecanone (**2e**). Applying the reaction to branched cycloalkanes leads to a mixture of various oxidation products, with general selectivity toward forming ketones rather than alcohols (see Supplementary Results 2.8).

The reactions for cycloalkenes 5a-e, portrayed in Fig. 2b, were carried out in 5 mL PTFE cells under 100 vol% oxygen atmosphere, since here the best result for cyclododecene (**5c**) oxidation was observed (Fig. 2b). Besides of suberic acid (**6b**, 46% yield) and dodecanedioic acid (**6c**, 78% yield) also adipic acid (**6a**) could be synthesized in 19%. As an example of a bicyclic substrate, norbornene led to the formation of 1,3-cyclopentane diacid (**6d**) with a 45% yield. In addition to the formation of α,ω-diacids from disubstituted cycloalkenes, α,ω-ketocarboxylic acids **6e** can be obtained from trisubstituted cycloalkenes. As an example of naturally occurring terpenes, (*S*)-

(−)-limonene could also be converted to the corresponding keto-carboxylic acid **6f**, with a yield of 17%, which is comparable to that of adipic acid (**6a**) from cyclohexene (**5a**).

Cycloalkenes of technical grade contain fully saturated analogs as impurities from industrial synthesis steps to specific percentages. Conversion of impure starting materials is beneficial for an industrial application if the impurities do not interfere with the target reaction. To test the applicability of this protocol, a co-electrolysis of **1e** and **5c** has been successfully performed, whereby the yields of ketone **2e** and dicarboxylic acid **6c** depend on the composition of the starting materials (Fig. 2c). For example, the yield of cyclododecanone (**2e**) increases with lower molar amounts of cyclododecane (**1e**) within the substrate composition, and vice versa. The same observation has been made for the unsaturated ring **5c** and the diacid **6c**. Despite that, both substrates can be specifically converted into the ketone or diacid in parallel. A further co-electrolysis example for cyclooctane (**1c**) and

cyclooctene (**5b**) is given in Supplementary Table 3. Here the received products show generally lower yields with 2–4% of cyclooctanone (**2c**) and 40–45% of suberic acid (**6b**).

Besides batch-type electrolysis, flow electrolysis methods are becoming increasingly popular due to the inherent advantages of continuous process control[57,58]. The adaptability of the presented method towards different cell and process designs is presented by conducting the dicarboxylic acid synthesis also in an electrochemical flow set-up (Fig. 2d). These experiments were carried out by pumping the electrolyte through the cell in a cyclic manner until a charge amount of 2–4 $F$ was applied[59]. A small-scale trial with cyclododecene (**5c**) in a concentration of 0.05 mol L$^{-1}$ in isobutyronitrile led to a dodecanedioic acid (**6c**) formation of 76%. For larger technical applications, the solvent was exchanged, and a scale-up was pursued. After a few variation trials within the same set-up (see Supplementary Table 4), the reaction could be optimized with a non-toxic dimethyl carbonate/isopropanol (DMC/*i*-PrOH) mixture and a substrate concentration of (0.5 mol L$^{-1}$) to yield **6c** in 52%.

With the presented method, not only cyclic alkenes **5** but also linear alkenes, here fatty acids **7**, can be converted to their corresponding acids (Fig. 2e). The reaction was performed with elaidic acid and oleic acid, as prominent examples for fatty acids, which only differ from their *E-Z* isomerism. Furthermore, the corresponding methyl oleate, a component of biodiesel, was used to investigate the stability of ester groups under the applied conditions. In all cases, the double bond cleavage successfully led to pelargonic acid (**8**), azelaic acid (**9b**), and mono-methyl azelate (**9a**), respectively, with yields between 33 and 46%.

## Mechanistic studies

Instrumental and wet-chemical experiments were carried out to uncover mechanistic detail, which is schematically illustrated in Fig. 3a–c. The initial electrochemical step is assigned to the nitrate oxidation, as shown via cyclic voltammetry experiments (Fig. 3d, e). The formation of nitrate radicals could be proved via a radical quenching experiment conducted on cyclooctene (**5b**) under an argon atmosphere to avoid an oxygen reduction reaction. Among several unidentified materials, the nitrated intermediates **10b** and **11b** were detected via GC-MS; their formation is illustrated by the mechanistic proposal shown in Fig. 3f. Further GC-MS analysis results are given in Supplementary Fig. 17. The conclusion that nitrate serves as a mediator due to its reformation after oxidation to a radical is supported by anion chromatography measurements. After electrolysis and an extraction workup, only nitrate remains in the aqueous layer (Fig. 3g). Furthermore, nitrate is not getting electrochemically reduced to nitrite as shown by an anion chromatography measurement (see Supplementary Fig. 15), and a negative Griess test (Fig. 3h). Instead, the cathodic counter reaction is provided by reduction of dissolved oxygen, that originates from the atmosphere above the reaction solution (Fig. 3i). The resulting superoxide radical anions that are presumably stabilized by lipophilic organic cations like TBA[60] are assumed to react with the hydrocarbon radicals to peroxide intermediates. A peroxide-specific detection test using titanyl sulfate[61,62] showed a positive result for these species directly after electrolysis, as a yellowish coloring of the specific peroxotitanyl ion $(TiO_2)^{2+}$ appears, comparable to the test with other hydroperoxides (Fig. 3j). The dissolved oxygen concentration at a partial pressure of 1 atm has been determined by cyclic voltammetry studies and applying the Randles-Ševčík equation (see Supplementary Note 3.1). At 100 vol% of O$_2$ in the reaction atmosphere, a concentration of $9.5 \pm 0.4$ mmol L$^{-1}$ was observed, comparable to literature values[63]. Furthermore, a steady pH value before and after the reaction of 5–6 was observed with standard pH indicator paper, implying no drift into an acidic or alkaline environment (see Supplementary Fig. 16). Carrying out electrolysis with cyclooctane (**1c**) substrate revealed an increased water content compared to the one without a

substrate after electrolysis (Fig. 3k). Measurements were conducted using a Karl Fischer titration method (see Supplementary Note 3.2.5). For the C=C double bond cleavage, an addition of the nitrate radical onto the double bond is proposed, according to the findings of the control experiment (Fig. 3f) and to literature descriptions[64]. Further reaction with a superoxide radical lead via an unknown pathway to the formation of aldehydes (Fig. 3b) that are obtained as possible intermediates regarding an HRMS analysis observation of the crude reaction mixture after electrolysis of **5c** to **6c** (see Supplementary Fig. 13). Afterwards, literature described autooxidation like mechanism from aldehydes to carboxylic acids is assumed to occur[65] (Fig. 3c). The oxidation of free carboxylic acids as a competing anodic reaction to nitrate oxidation was not observed (see Supplementary Fig. 18c). The possibility of cathodic hydrogen formation, which is a potential hazard when combined with oxygen, was considered unlikely based on the aprotic conditions applied for the reactions and cyclic voltammetry experiments on the reductive behavior of the electrolyte systems (see Supplementary Note 3.2.7). Dissolved oxygen is preferentially reduced in the applied electrolyte systems over protic solvent additives such as 2-propanol and water. Studies on this method's potential large-scale application are subject of further investigation.

## Further method application

To demonstrate the broad applicability of the presented method, the substrate scope was successfully extended to toluene substrates **12**. Benzylic oxidation to either the benzaldehydes **13'**, including subsequent derivatization with semicarbazide hydrochloride **15** to the corresponding semicarbazones **13**, or the benzoic acids **14** could be promoted selectively via tuning the reaction conditions (Fig. 4a). Reaction monitoring via $^1$H NMR spectra shows the aldehyde as an intermediate, which is further converted to the corresponding carboxylic acid when the toluene substrate is almost entirely converted (Fig. 4b). No intermediates containing an alcohol group were observed. Optimization reactions were carried out in undivided 5 mL PTFE cells (see Supplementary Table 5), while a 25 mL beaker-type glass cell was used for the scale-up reactions. Suitable conditions for the aldehyde formation differ significantly from those for the acid formation by changing the current density, the charge quantity, and the substrate concentration. By applying conditions of 10 mA cm$^{-2}$, 5 $F$, and 0.02 mol L$^{-1}$ of **12**, the reaction can be stopped after the selective formation of the benzaldehyde **13'**. When these parameters are increased to 30 mA cm$^{-2}$, 7–12 $F$, and 0.1 mol L$^{-1}$ of **12**, the formation of benzoic acid **14** can be promoted. Mainly methylated toluenes (xylenes) were chosen as substrates, to investigate over-oxidation reactions at the remaining benzylic positions. Only one methyl group is selectively oxidized to the aldehyde and then further into the carboxylic acid. As by-products of the benzoic acid synthesis, *N*-acetylbenzamides **14'** were obtained in a 3–4% yield range. Presumably, acyl radicals are formed as described in Fig. 3c, which react with acetonitrile as a radical scavenger and are further oxidized to **14'** (see Supplementary Fig. 19). Derivatization of the aldehydes to semicarbazones **13** was carried out on the one hand, to prevent further autooxidation of the aldehyde and therefore falsification of the yield determination, and on the other hand to provide a simple protocol for semicarbazone **13** synthesis. Semicarbazones show pharmacological versatility and are known for their anticonvulsant[66,67] and potential anticancer[68] properties.

## Discussion

In conclusion, one simple, sustainable approach was developed to facilitate two challenging electrochemical oxidation reactions. With the presented method, cyclic alkanes and (cyclic) alkenes can be oxidized into ketones and (di)carboxylic acids, respectively. The overall moderate yields are balanced by the resource and material saving advantages, like the dual role usage of supporting electrolyte and

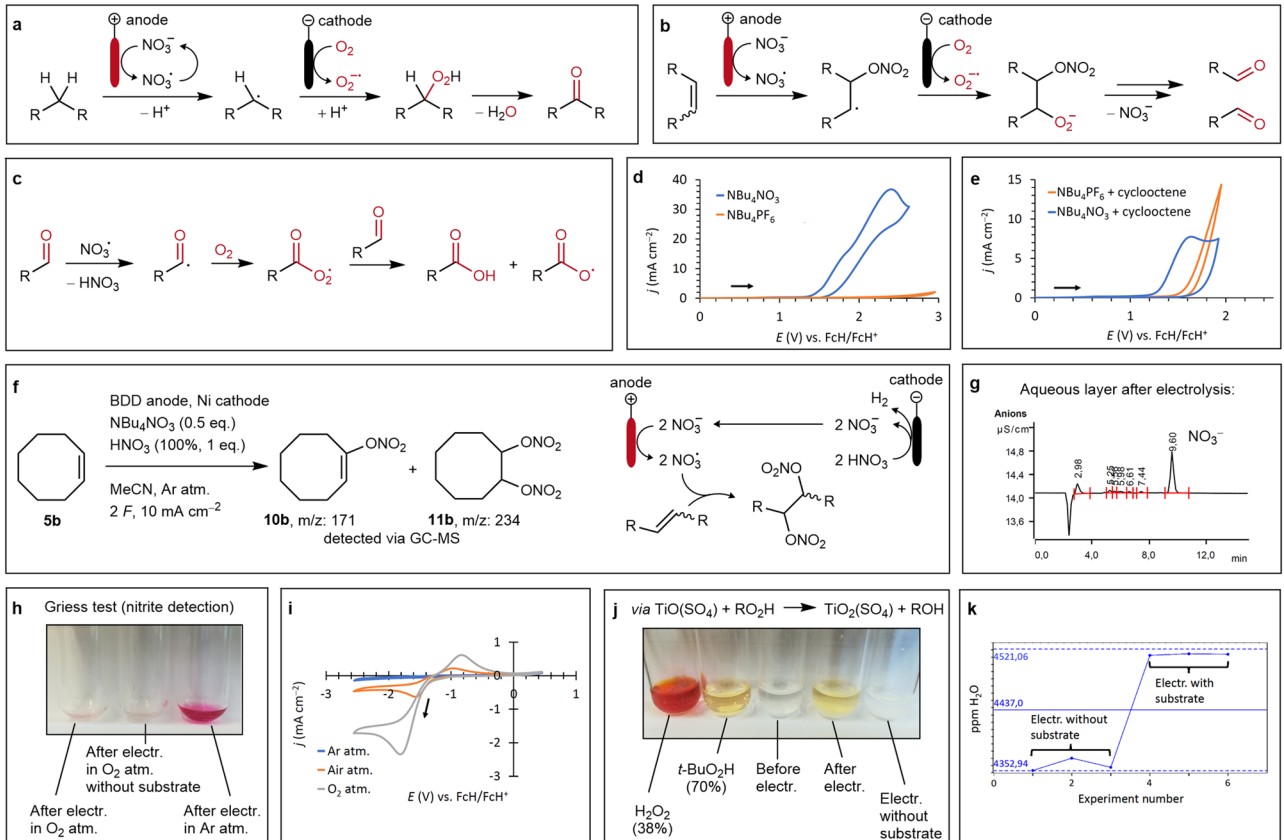

**Fig. 3 | Mechanistic studies and proposal.** atm.: atmosphere, electr.: electrolysis. Oxygen as $O_2$ source and oxo-groups are labeled in red. **a** Schematic, mechanistic proposal for cycloalkane oxidation. **b** Schematic, mechanistic proposal for alkene double bond cleavage. **c** Schematic, mechanistic proposal for further oxidation of aldehyde species to carboxylic acids. **d** Cyclic voltammetry of nitrate anion oxidation compared with $PF_6^-$. Electrolyte: acetonitrile (5 mL), $NBu_4NO_3$ (0.1 mol L$^{-1}$), or $NBu_4PF_6$ (0.1 mol L$^{-1}$). Conditions: glassy carbon disk (working electrode, 3 mm diameter), glassy carbon rod (counter electrode), Ag/AgCl in saturated LiCl/EtOH (reference electrode), Ferrocene/Ferrocenium (FcH/FcH$^+$) as internal reference ($E_{1/2} = 0.55$–$0.57$ V), 50 mV s$^{-1}$. **e** Cyclic voltammetry of nitrate anion oxidation compared with $PF_6^-$ in the presence of cyclooctene **5b**. Electrolyte: see Fig. 3d, **5b** (0.2 mol L$^{-1}$). Conditions: see Fig. 3d, ($E_{1/2}$(FcH/FcH$^+$) = 0.55–0.58 V). **f** Quenching experiment in an undivided 5 mL PTFE and schematic mechanism proposal for the formation of nitrated species. MeCN: acetonitrile. For further information, see Supplementary Note 3.2.6. **g** Anion chromatography of the aqueous layer after extractive workup. For preparative information, see Methods: Workup procedure for cyclic ketones and benzaldehydes. **h** Griess test for nitrite detection is negative if the reaction solution is exposed to an $O_2$ atmosphere (100 vol%) and positive under argon. See Supplementary Note 3.2.1 for detailed protocol. **i** Cyclic voltammetry of oxygen reduction in argon (blue line), air (orange line), and oxygen (gray line) atmosphere above the electrolyte solution. Electrolyte: acetonitrile (5 mL), $NBu_4NO_3$ (0.1 mol L$^{-1}$). Conditions: see Fig. 3d, ($E_{1/2}$(FcH/FcH$^+$) = 0.55 V). **j** Peroxide test with titanyl sulfate. See Supplementary Note 3.2.4 for detailed protocol. **k** Karl Fischer titration after electrolysis of the reaction solution. See Supplementary Note 3.2.5 for detailed protocol.

electrochemical mediator. Atmospheric oxygen is crucial for the reaction's success, making additional oxidizing agents redundant. No transition metal use is needed for the procedure, as even the electrode material is based on simple glassy carbon. Furthermore, the method shows widely applicable oxo-functionalization properties, as an application for benzylic oxidation successfully led to aldehydes and benzoic acids. This work provides a valuable contribution to sustainable, preparative chemical processes and allows the production of industrially relevant chemicals for large-scale technical plastics production.

## Methods
Generated data on experimental procedures can be retrieved from this section and the Supplementary Information.

### Electrolysis in 5 mL PTFE cells (GP 1)
The electrolysis set-up is commercially available as IKA Screening System from IKA-Werke GmbH & Co. KG, Staufen, Germany. Gas distributor and associated caps can be purchased separately from IKA. In a 5 mL undivided PFTE cell with a magnetic stirrer, the substrate (0.25–1 mmol) and the supporting electrolyte ($NBu_4NO_3$, 0.5 eq.) are dissolved in acetonitrile or isobutyronitrile (5 mL). The cell is equipped with a cap, including a gas adapter, where two glassy carbon electrodes (7 cm × 1 cm × 0.3 cm) are fixed at a distance of 0.5 cm from each other. The immersed electrode surface is 1.8 cm$^2$. The cell is fixed in a stainless-steel set-up, and while stirring, the atmosphere within the cell is filled with oxygen gas (2.5 technical grade, purity ≥99.5%). During electrolysis, a constant flow of oxygen of 20 mL min$^{-1}$ into the cell is ensured. Constant current electrolysis with a current density of 5–10 mA cm$^{-2}$ and an applied charge of 4–8 F is performed. For set-up information, see Supplementary Fig. 1.

### Electrolysis in a 100 mL round-bottom flask (GP 2)
In a 100 mL three-necked round-bottom flask with an NS29 Teflon plug incl., electrode holders, magnetic stirrer, and bubble counter, the substrate (5 mmol) and supporting electrolyte ($NBu_4NO_3$, 0.5 eq.) are dissolved in acetonitrile or isobutyronitrile (25 mL). The cell is equipped with two glassy carbon electrodes (3 cm × 1 cm × 0.3 cm), fixed at a distance of 0.5 cm from each other. The immersed electrode surface is 1.3 cm$^2$. While stirring, the atmosphere within the cell is filled with oxygen gas (2.5 technical grade, purity ≥99.5%). During electrolysis, a constant flow of oxygen of 20 mL min$^{-1}$ into the cell is ensured.

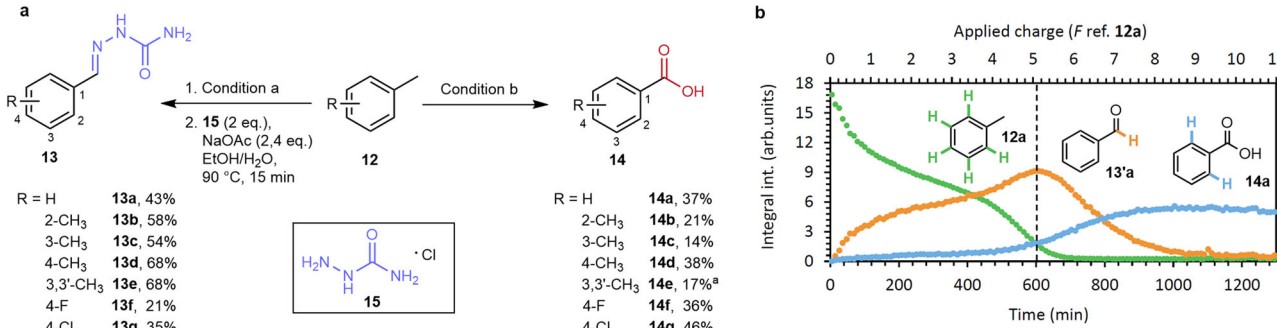

**Fig. 4 | Benzylic oxidation as a further application of the method. a** Oxidation of toluene substrates to either semicarbazones **13** (semicarbazide moiety in blue) via benzaldehydes or to benzoic acids **14** (carboxyl group in red). Yields are isolated. Condition a: undivided 25 mL beaker-type cell, glassy carbon anode, and cathode, acetonitrile (25 mL), NBu$_4$NO$_3$ (1 eq.), substrate (0.02 mol L$^{-1}$), 33 °C, O$_2$ atmosphere (100 vol%), 400 rpm, 5 $F$, 10 mA cm$^{-2}$. Condition b: undivided 25 mL beaker-type cell, glassy carbon anode, and cathode, acetonitrile (25 mL), NBu$_4$NO$_3$ (0.5 eq.), substrate (0.1 mol L$^{-1}$), 27 °C, O$_2$ atmosphere (100 vol%), 500 rpm, 12 $F$, 30 mA cm$^{-2}$. $^a$7 $F$. **b** Reaction monitoring via a 60 MHz NMR benchtop spectrometer. The $^1$H integral intensity was normalized to one proton for better comparison. Recorded tracks of the corresponding protons are assigned by colors (green dots: aromatic protons of **12a**, orange dots: aldehydic proton of **13'a**, blue dots: aromatic protons in the *ortho* position of **14a**).

Constant current electrolysis with a current density of 10 mA cm$^{-2}$ and an applied charge of 4–8 $F$ is performed at 20–30 °C. For set-up information, see Supplementary Fig. 2.

### Electrolysis in 25 mL beaker-type glass cells (GP 3)
The cells are commercially available as SynLectro™ Electrolysis Platform from Sigma-Aldrich (Merck KGaA, Darmstadt, Germany). In a 25 mL beaker-type glass cell with gas inlet attachment, an NS45/40 Teflon plug incl., electrode holders, and magnetic stirrer, the substrate (0.5–5 mmol) and supporting electrolyte (NBu$_4$NO$_3$, 0.5–1 eq.) are dissolved in acetonitrile (25 mL). The cell is equipped with two glassy carbon electrodes (7 cm × 1 cm × 0.3 cm), fixed at a distance of 0.5 cm from each other. The immersed electrode surface is 4.5 cm². While stirring, the atmosphere within the cell is filled with oxygen gas (2.5 technical grade, purity ≥99.5%). During electrolysis, a constant flow of oxygen of 20 mL min$^{-1}$ into the cell is ensured. Constant current electrolysis with a current density of 10–30 mA cm$^{-2}$ and an applied charge of 5–13 $F$ is performed at 27–33 °C. For set-up information, see Supplementary Fig. 3.

### Electrolysis in a flow cell (GP 4)
The electrolysis set-up is made in the university's machine shop[69,70] and is commercially available as IKA ElectraSyn flow. A peristaltic Ismatec Reglo ICC with a TYGON 2765-175 (ID 2.06 mm) hose was used as pumping system. The undivided flow cell was equipped with glassy carbon electrodes (2 × 6 cm²). In a 20 mL snap cap vial, the substrate (0.5–5 mmol) and supporting electrolyte (NBu$_4$NO$_3$, 0.5–1 eq.) are dissolved in isobutyronitrile or a dimethyl carbonate/isopropanol mixture (10 mL). The reaction solution is pumped with 10–18 mL min$^{-1}$ through a t-piece, where oxygen gas is passed through with 10–20 mL min$^{-1}$. The combined, segmented flow is pumped through the flow cell (electrode gap: 0.05 cm) back into the snap cap vial in a cycling operation mode. Constant current electrolysis with a current density of 5–20 mA cm$^{-2}$ with an applied charge of 2–4 $F$ is performed at 20–50 °C, whereby the reservoir is exposed to the heating. During electrolysis, a constant flow of oxygen into the t-piece is ensured. For set-up information, see Supplementary Fig. 4.

### Workup protocol for (di)carboxylic acids and benzoic acids
After electrolysis, the solvent is recovered by distillation under reduced pressure. The low-boiling starting material leftover can be recovered by extraction with n-pentane out of the nitrile distillate and monitored via thin-layer chromatography (TLC). The distillation residue is dissolved in ethyl acetate (10 mL) and extracted with HCl aq.

(0.1 mol L$^{-1}$, 10 mL) or HNO$_3$ aq. (0.1 mol L$^{-1}$, 10 mL, for recycling of the supporting electrolyte), whereby the supporting electrolyte remains in the aqueous layer and can be recovered quantitatively. The organic fraction is reduced in vacuo to yield the crude product. For purification, the residue is washed with NaOH aq. (1 mol L$^{-1}$, 10 mL) and extracted with diethyl ether or n-pentane (10 mL), monitored via TLC. After dropwise acidification of the aqueous layer with conc. HCl aq. to pH 1 and extraction with ethyl acetate (2 × 10 mL), the combined organic layers are dried over MgSO$_4$ and reduced in vacuo to yield the desired product.

### Workup protocol for cyclic ketones and benzaldehydes
After electrolysis, the solvent is recovered by distillation under reduced pressure. The low-boiling starting material leftover can be recovered by extraction with n-pentane out of the nitrile distillate. The distillation residue is extracted with water (20 mL) and cyclohexane (20 mL) or diethyl ether (20 mL), whereby the supporting electrolyte remains in the aqueous layer and can be recovered quantitatively. The organic layer is dried over MgSO$_4$ and reduced in vacuo to yield the desired product. Special attention has to be paid to the distillations since short-chained cyclic ketones are volatile. Solvent evaporation was carried out to a maximum of 250 mbar at 50 °C for cyclohexane or 970 mbar at 40 °C for diethyl ether.

For derivatization of the benzaldehydes to the semicarbazones: After extraction and evaporation of the solvent from the dried organic layer, the residue is immediately dissolved in 3 mL abs. ethanol and is added dropwise to a 90 °C preheated solution of semicarbazide hydrochloride (2 eq.) and sodium acetate trihydrate (2.4 eq.) in 10 mL water. Heating and stirring are continued for 15 min. Afterward, the solvents are evaporated, and the precipitating semicarbazone is suspended in cold, deionized water, filtrated, washed with cold water, and dried in vacuo.

### Workup protocol for the co-electrolysis
After electrolysis, 10 mg of 1,3,5-trimethoxybenze as an internal standard was added to the reaction solution. Three drops were eluted with ethyl acetate through ~330 mg silica 60 M and filled into a vial for quantitative GC-FID analysis. Before the analysis, an external GC calibration of the ketone was performed (see Supplementary Fig. 6). The dicarboxylic acid has been isolated regarding the following workup: The solvent is recovered by distillation under reduced pressure. The low-boiling starting material leftover can be recovered by extraction with n-pentane out of the nitrile distillate and monitored via thin-layer chromatography (TLC). The distillation residue is dissolved in ethyl

acetate (10 mL) and extracted with HCl aq. (0.1 mol L$^{-1}$, 10 mL), whereby the supporting electrolyte remains in the aqueous layer. The organic fraction is reduced in vacuo. For purification, the residue is washed with NaOH aq. (1 mol L$^{-1}$, 10 mL) and extracted with diethyl ether or n-pentane (10 mL), monitored via TLC. After dropwise acidification of the aqueous layer with conc. HCl aq. to pH 1 and extraction with ethyl acetate (2 × 10 mL), the combined organic layers are dried over MgSO$_4$ and reduced in vacuo to yield the dicarboxylic acid product.

### Workup procedure for carboxylic acids from fatty acids
After electrolysis, 50.5 µL propionic acid was added to the reaction solution as an internal standard. Three drops were eluted with ethyl acetate through ~330 mg silica 60 M and filled into a vial for quantitative GC-FID analysis. Prior to the analysis, an external calibration of the mono-carboxylic acid was performed (see Supplementary Fig. 7). The dicarboxylic acid was isolated via column chromatography using silica 60 M (cyclohexane/ethyl acetate = 1:1 with 1 vol% glacial acetic acid).

## Data availability
The authors declare that all data supporting the findings of this study are available within the paper and its Supplementary information. Source data are provided with this paper.

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

## Acknowledgements

S.R.W. gratefully acknowledge the financial support by Forschungsinitiative Rheinland-Pfalz in the frame of SusInnoScience. J.N. thanks J.L. Schröder for preparative support, A.L. Rauen for initial guidance in the subject area and helpful discussions, S. Hofmann for helpful discussions and invention disclosure support, S. Arndt for the introduction in ion chromatography measurements and J. Herszman for introduction into the coulometric titration apparatus.

## Author contributions
J.N., F.W., F.-E.B., and S.R.W. conceived this work and designed the experiments. J.N., K.H., S.M., I.C.M., and D.M. conducted the experiments and analyzed the data. F.W. and F.-E.B. discussed the results and commented on the manuscript. J.N. and S.R.W. wrote the manuscript and the Supplementary information.

## Funding

## Competing interests

F.W. is an employee at Evonik and holds shares in the company. F.-E.B. was an employee at Evonik and is retired. S.R.W., F.W., F.-E.B., J.N., and K.H. are inventors in patent applications regarding the manuscript aspects of cyclic alkane oxidation to ketones, cyclic alkene oxidation to dicarboxylic acids, co-electrolysis of cyclic alkanes and alkenes, and oxidative fatty acid cleavage. The patent applications are filed at the European Patent Office and have not yet been published. Searchable application numbers are available upon publication. The remaining authors declare no competing interests.
