## [Peer Review File · Nature Communications]

REVIEWER COMMENTS

Reviewer #1 (Remarks to the Author):

The authors describe some interesting, novel, and exciting chemistry. It is rendered particularly attractive because it uses nitrate salts in the dual role of supporting electrolyte and anodic mediator. Cathodic reduction of molecular oxygen ensures the incorporation and use of both of the electrode processes. The overall yields range from poor to moderate, though considering the important and diverse set of transformations that are possible, the low yields can be overlooked. The mechanistic proposals appear to be sound, being supported using both old-fashioned wet chemistry as well as instrumental techniques (MS, etc.). In addition, and as the authors write: "This work provides a valuable contribution to sustainable, preparative chemical processes, and allows the production of industrial relevant chemicals for large-scale technical plastics production."

Despite these positive attributes, I cannot recommend publication at this time. There are too many grammatical concerns, particularly important being those that lead to lack of clarity regarding meaning. These issues are easy to fix. Below, I offer potential solutions to many of the issues. I hope they are useful.

- Use "allows rapid access to valuable ..." rather than: "bonds allows a fast, synthetic access to valuable"
- Use "unactivated" rather than "inactivated" in: "inactivated methylene"
- Recommend using "Comparable to those previously mentioned" rather than "Comparable to the before mentioned ones, these usually"
- Delete "ones" from: "the most frequently described ones. Older protocols"
- Call me old-fashioned, but when it comes to C=C cleavage, I'll choose ozone every time. Sure, there are dangers. However, with proper precautions, it's clean, efficient, and safe.

- “have scarcely been described” rather than: “oxidation of C(sp³)-H bonds to ketones are barely described”
- Suggest writing “Just as with alkanes, only a few ...” rather than: “Similarly to the alkane oxidation, a few electrochemical methods for the double bond cleavage to dicarboxylic acids are known”
- Recommend “greater than a stoichiometric amount ...” rather than “over-stoichiometric amounts of oxidisers leads to reagent waste.”
- Delete “getting” from “use consisting of the solvent and the supporting electrolyte salt, is getting oxidised prior to these substrates”
- Suggest using “Frequently, hydrocarbons display poor solubility in these solvents” rather than: “hydrocarbons showing often solubility issues in these solvents
- Use “Inspired by literature, we focused upon the use of nitrate salts ...” rather than: “Inspired by literature, the focus was set to nitrate salts as supporting electrolyte and electrochemical mediator in a dual role... “
- The following is not as clear as it should be. I can imagine, for example, several ways to “place” air above a reaction solution. Please be more specific. I am referring to: “were conducted under ambient conditions with air above the reaction solution.”
- Where’s the rest of the mass? Why is the mass balance so low? See, for example: “However, a maximum yield of 31% was obtained for 2c under 20 vol.% O₂ and 10 m”
- Change “The agitation speed of the stirrer within the cell, has a significant influence” to read: “The rate of stirring has a significant influence”
- Be consistent. Sometimes “splitting” is used, other times it’s not. Perhaps “cleavage” would be more appropriate? See “The C=C double bond splitting of cyclic alkenes 5 (Fig. 3) has been discovered by applying the same” and also “mechanistic proposal for alkene double bond splitting”

- Suggest modifying “Following this result, other different membered rings from six to twelve methylene groups were tested in this set-up (Fig. 3a)” to read: “Next, we used this setup to explore the oxidation of cyclic alkanes containing six to twelve methylene groups (Fig. 3a).”
- Change “with” to “in” – “could be synthesised with 19%. As”
- Use “The resulting superoxide radical anion” rather than: “The hereby formed superoxide radicals,”
- Change “happen” to “occur” in: “aldehydes to carboxylic acids is assumed to happen”
- Rather than writing “different atmospheres” be specific. Fig. 4b shows “Ar atm”, “air atm”, and “O2 atm”. Use these labels as “different atmospheres” is unnecessarily vague.
- Change “addressed” to “attributed” in: “This circumstance is addressed to a disturbed O2 adsorption on the electrode at”
- Suggest modifying “The experiment has been carried out in a cycling mode, pumping the electrolyte multiple times through the cell into a reservoir, until ...” To read: “These experiments were carried out by recycling the electrolyte through the cell until”
- What is meant by “applied charge amount passed through”? Please be more specific.
- Specify what is meant by “O2 atmosphere”. It seems to me that there are a number of different scenarios that could be described as being under an O2 atmosphere.
- Modify “The reactions for the substrate variation of the cycloalkenes were carried out in 5 mL PTFE cells under” to read: “The reactions for cycloalkenes 5a-e portrayed in Fig. 3b, were carried out in 5 mL PTFE cells under”
- Suggest modifying “Besides the synthesis of diacids from disubstituted double bonds, trisubstituted ones lead to formation of α,ω -ketocarboxylic acids, as shown on the example 6e” to

read: "In addition to the formation of α,ω -diacids from disubstituted cycloalkenes, α,ω -ketocarboxylic acids 6e can be obtained from trisubstituted alkenes."

- Suggest modification of "To investigate mechanistical correlations of the cycloalkane (Fig. 4f) and alkene (Fig. 4h and 4i) oxidation reactions, several experiments have been conducted" to read: "To uncover mechanistic detail, the experiments illustrated in Figures 4f, h, and i were carried out."
- Modify "atmosphere, whereby among others the nitrated intermediates (10b) and (11b) were detected via GC-MS and illustrated by a mechanistic proposal (Fig. 4k)" to read: "atmosphere, whereby among several unidentified materials, the nitrated intermediates (10b) and (11b) were detected via GC-MS; their formation is illustrated by the mechanistic proposal shown in Fig. 4k."
- Suggest modifying "The C=C double bond splitting of cyclic alkenes 5 (Fig. 3) has been discovered by applying the same methodology" to read: "Application of the same methodology to cyclic alkenes 5 revealed that they undergo C=C double bond cleavage as shown in Fig. 3."
- Suggest modifying "Following the observation via HPLC-MS of a dicarboxylic acid 6 formation, the reaction conditions were varied using cyclooctene (5b) as model substrate" to read: "Following the observation of a dicarboxylic acid 6 via HPLC-MS, the reaction conditions were varied using cyclooctene (5b) a as model substrate"

The following is a list of other portions of the text that require work/rewriting/clarification. I leave it to the authors to do so.

Adversely, the method is barely suitable for large technical applications as 1,1,1,3,3,3-hexafluoropropan-2-ol (HFIP) is used as solvent, which global warming potential (GWP100) is approximately

Electrolysis was carried out with 4 F regarding 1c and 10 mA/cm² leading to a yield

the ketone yields, compared to the increasing ring sizes, is approximately following the transannular

diacid 6c. Despite of that, both substrates can be converted by their own target reactions parallel to each other.

nitrate radicals could be indirectly observed due to a control experiment, conducted

This observation was supported by anion chromatography, as solely nitrate remains in the aqueous layer subsequently to an extractive work up after the electrolysis (Fig. 4a).

an increased water content, if the sample contained the hydrocarbon substrate (Fig. 4d).

For the presented method, a further application field has been demonstrated

Reaction monitoring via ^1H NMR spectra recording shows the aldehyde as

With lower conditions around 10 mA/cm^2 , 5 F, and 0.02 mol/L of 12

aldehyde 13', while increasing these conditions to 30 mA/cm^2 , 7–12 F and

A plausible explanation leads via the formation of acyl radicals (as postulated in Figure 4),

Reviewer #2 (Remarks to the Author):

Waldvogel and co-workers reported the electrochemical valorization of simple alkanes and cycloalkenes into valuable chemicals. The key of the present system is the combined use of NO_3^- as a mediator and dioxygen gas, where NO_3^- serves as a HAT reagent and O_2 gas enables oxygen functionality. Under the optimized reaction condition, cycloalkanes afforded cyclic ketones, whereas cyclic alkenes afforded dicarboxylated ring-opening products, and product selectivities of these

reactions are generally quite high. The authors also demonstrated the successful application of this method to linear fatty acids. NO_3^- is used both as a mediator and the electrolyte, and importantly, they demonstrated the recovery of this salt after the reaction, demonstrating the potential high sustainability of the present system.

The present work represents a promising strategy for the sustainable valorization of hydrocarbons, which is still a significant challenge even with state-of-the-art technologies. The yields are not necessarily high (78% in maximum, and mostly $\sim 50\%$), but that in no way diminishes the value of this study, given the difficulty of the reactions that challenged the authors. Thus, the reviewer supports the publication of the present study in Nature Communications, after addressing the point listed below.

Scientific points

- In the abstract, the authors claim “This resource-friendly method is based on nitrate salts in a dual role as anodic mediator and supporting electrolyte, which can be fully recovered and recycled”. However, the manuscript and SI (supporting data 2.7) only demonstrate the recovery, and there is no data of recycle. Is the collected nitrate salt from the reaction mixture still usable?
- This system relies on the convergent paired electrosynthesis via the oxidation of nitrate anion and the reduction of oxygen gas. The generation of H_2 gas at the cathode as a by-product (especially when using protic solvent) potentially causes an explosion by the co-existence of O_2 . This point would be especially crucial when directing large-scale applications for industrial processes. The authors should give some data or discussion in the manuscript.
- Related to the question above, if there is no competing HER, the reaction media becomes acidic in the late stage of the reaction. The reviewer assumes the acidic condition changes the reactivity of O_2^- . Are there any data or comments on this point?
- Carboxylic acids generated via the cleavage of alkenes seem to be easily oxidized under electrochemical conditions. Does the oxidation of carboxylic acid compete with the oxidation of nitrate?
- There is a unique contrast that the olefin cleavage reaction solely afforded dicarboxylic acids, whereas the oxidation of toluenes followed sequential oxidation to benzoic acid via benzaldehyde. Why no aldehyde intermediate was obtained during the olefin cleavage reaction, despite the proposed mechanism in Fig. 4i proceeding via the formation of an aldehyde intermediate?
- To pursue large-scale production, the development of a flow system would be the key, and the authors demonstrated promising results (Fig. 3d). The reviewer is wondering how oxygen gas is reacting in the flow cell. As explained in the manuscript (experimental section, GP4), the flow reaction media is segmented. Considering the very small volume of the electrochemical flow cell used herein (electrode gap: 0.050 cm), the introduction of gas seems to be crucial. But in the reality, the reaction proceeds well under the slightly modified conditions from the batch system. Can the authors explain why the segmentation by O_2 gas did not ruin the reaction?

– Is there any product formed in the absence of O₂ (Fig. 2, Entry 3)? The reviewer is curious if a desaturation reaction such as olefin formation occurs if there is no O₂.

Non-scientific points

– Figures in the manuscript should appear following alphabetical order. For example, Fig. 4k appears first in the manuscript before mentioning Figs. 4a-j. It makes the manuscript hard to read. Context and Figures should be organized.

– Line 32, a comma is missing after “...application fields...”.

– Line 83 should be “oxo-functionalization”, not “oxo-functionalisation”

– Line 114, a comma is missing after “cell...”.

– Line 115, “showing” should be “show”.

– In line 119 (... Inspired by literature, ...), it is unclear which literature the authors indicate.

Reviewer #3 (Remarks to the Author):

Authors describe the use of nitrate salts with a dual role as catalyst and electrolyte of these reactions. The oxidation of alkanes is a challenging transformation due to the lack of activating groups in these molecules. On the other hand, the selective oxidative cleavage of alkenes provides high yields of the corresponding diacids.

Apart from this comments, I have a few more curiosities:

- I miss a couple of examples of branched alkanes in the first part. Would the tertiary carbons get oxidized easier and selectively? Would the alcohols be obtained as sole products? Trying this reaction with a couple of more complicated alkanes would be illustrative. The same in the oxidative cleavage of alkenes. Would a more substituted cyclic alkene give the corresponding hydroxy acid?

- In the mechanism of the oxidation of toluene derivatives, aldehyde is formed as intermediate, which afterwards gives the acid. Is this not observed in the case of the alkene oxidative cleavage?

- The authors say that ‘Inspired by literature’ (line 119) but no reference is mentioned here.

- Why is a lower temperature (5°C) beneficial for the reaction?

- Authors tried using BDD and graphite electrodes as well, but no combination of them was tested. Sometimes, a different material in anode and cathode can positively affect to the reaction performance.
- For the oxidation o alkanes, the optimized conditions were found using 20 vol% oxygen. However, the reaction is performed with pure oxygen atmosphere, and this could be due to 'larger contact area between the oxygen containing atmosphere and the reaction solution. Was it re-tested using air or 20 vol% in O₂?
- Figure 4: this reviewer suggests a better organization of the panels of this figure. The text and the figure have a different organization and, since the manuscript deals with two reactions, reading this mechanistic part with the panels unorganizaed regarding the text, is a bit conussing.
- In the case of running the reaction under pure N₂, no desired products are observed. Do the starting materials remain untouched?
- Authors mention (line 250) that 'in the case o cycloalkanes, the nitrate radicals serve as mediator, as the anion is reformed'. 0.5 eq of this nitrate salt are used, but yields are always below 50%, which means that the TON of this mediator is lower than 1. Is starting material remained untouched in this reaction? In this regard, authors mention that 'The reaction proved itself to be outstandingly selective, as only cyclooctanol (3c) and cyclooctane-1,4-dione (4c) were obtained as significant by-products with yields of 2% and 3%.' (lines 128-129). However, only 23% yield is obtained. What happens with the rest of material? If no starting material is recovered, I would not afirm that the method has an 'outstanding' selectivity of the method.
- Authors claim that the electrolyte can be recovered and recycled, but no recycling was tested.
- In the obtention of carboxylic acid, authors observe dehomologation only in the case of 6e. Was it not observed in any other case? Dehomologation of acids is common under electrochemical conditions due to overoxidation.
- Supplementary Figure 8: indicate the space of time between each NMR shown. Indicate that the time increases going up in the NMRs.

The authors thank all reviewers for their valuable comments, advice, and suggestions for improving the draft. In addition to the point-by-point response, general linguistic improvements were made in the manuscript to meet the high scientific standards of Nature Communications. All changes to the submitted manuscript have been marked in yellow.

REVIEWER COMMENTS

Reviewer #1 (Remarks to the Author):

The authors describe some interesting, novel, and exciting chemistry. It is rendered particularly attractive because it uses nitrate salts in the dual role of supporting electrolyte and anodic mediator. Cathodic reduction of molecular oxygen ensures the incorporation and use of both of the electrode processes. The overall yields range from poor to moderate, though considering the important and diverse set of transformations that are possible, the low yields can be overlooked. The mechanistic proposals appear to be sound, being supported using both old-fashioned wet chemistry as well as instrumental techniques (MS, etc.). In addition, and as the authors write: "This work provides a valuable contribution to sustainable, preparative chemical processes, and allows the production of industrial relevant chemicals for large-scale technical plastics production."

Our response: The summary is precise and sheds a positive light on our work; thank you. In particular, the emphasis on these important chemical transformations is very appropriate and in line with the core statement that the authors want to convey.

Despite these positive attributes, I cannot recommend publication at this time. There are too many grammatical concerns, particularly important being those that lead to lack of clarity regarding meaning. These issues are easy to fix. Below, I offer potential solutions to many of the issues. I hope they are useful.

Our response: The authors thank the reviewer for the valuable suggestions, which improved the document's readability and clarity.

1. Use "allows rapid access to valuable ..." rather than: "bonds allows a fast, synthetic access to valuable"

Our response: The sentence has been rephrased accordingly.

2. Use "unactivated" rather than "inactivated" in: "inactivated methylene"

Our response: The word "inactivated" has been exchanged with "non-activated".

3. Recommend using "Comparable to those previously mentioned" rather than "Comparable to the before mentioned ones, these usually"

Our response: The sentence has been rephrased accordingly.

4. Delete "ones" from: "the most frequently described ones. Older protocols"

Our response: The word has been deleted accordingly.

5. Call me old-fashioned, but when it comes to C=C cleavage, I'll choose ozone every time. Sure, there are dangers. However, with proper precautions, it's clean, efficient, and safe.

Our response: The authors agree with the reviewer that ozonolysis is a clean and efficient method for C=C double bond cleavage. However, ozonides decompose rapidly, so the reaction works exceptionally

well at low temperatures ($-78\text{ }^{\circ}\text{C}$ or even below)¹. Furthermore, the purchase of an ozone generator is necessary to produce ozone from oxygen. With the method reported here, oxygen can be applied directly. Finally, the energy required to generate ozone is quite high.

6. “have scarcely been described” rather than: “oxidation of $\text{C}(\text{sp}^3)\text{-H}$ bonds to ketones are barely described”

Our response: The sentence has been rephrased accordingly.

7. Suggest writing “Just as with alkanes, only a few ...” rather than: “Similarly to the alkane oxidation, a few electrochemical methods for the double bond cleavage to dicarboxylic acids are known”

Our response: The sentence has been rephrased accordingly.

8. Recommend “greater than a stoichiometric amount ...” rather than “over-stoichiometric amounts of oxidisers leads to reagent waste.”

Our response: The sentence has been rephrased accordingly.

9. Delete “getting” from “use consisting of the solvent and the supporting electrolyte salt, is getting oxidised prior to these substrates”

Our response: The word has been deleted accordingly.

10. Suggest using “Frequently, hydrocarbons display poor solubility in these solvents” rather than: “hydrocarbons showing often solubility issues in these solvents

Our response: The sentence has been rephrased: “Very lipophilic substrates like saturated hydrocarbons frequently display poor solubility in these solvents.” At this point, the authors want to highlight the lipophilicity of the substrates as their characteristic, as it provides explanatory support for the context.

11. Use “Inspired by literature, we focused upon the use of nitrate salts ...” rather than: “Inspired by literature, the focus was set to nitrate salts as supporting electrolyte and electrochemical mediator in a dual role... “

Our response: The sentence has been rephrased accordingly.

12. The following is not as clear as it should be. I can imagine, for example, several ways to “place” air above a reaction solution. Please be more specific. I am referring to: “were conducted under ambient conditions with air above the reaction solution.”

Our response: Thanks for the comment. The statement has been reworded and expanded for clarification: “The first experiments were conducted with ambient air in the gas space above the reaction solution. Due to the manufacturing design, the lid used for the electrochemical cell ensures an air exchange with the environment.”

13. Where’s the rest of the mass? Why is the mass balance so low? See, for example: “However, a maximum yield of 31% was obtained for 2c under 20 vol.% O_2 and 10 m”

Our response: The reviewer emphasizes a valid question. The reactivity of cycloalkanes for ketone synthesis is generally lower than that for diacid synthesis from cycloalkenes. After the reaction, significant amounts of unreacted cycloalkane are observed, usually in a range of 10–30% for 20 vol.% O_2 and 30–50% for 100 vol.% O_2 . To illustrate an example, the corresponding gas chromatogram and ^1H NMR spectrum of the reaction mixture presented in Fig. 2, entry 4 in the manuscript, are shown in Figs. 1 and 2 (see below).

Fig. 1: Gas chromatogram after electrolysis of cyclooctane (**1c**) to cyclooctanone (**2c**). Electrolysis conditions: undivided 5 mL PTFE cell, glassy carbon electrodes, acetonitrile (5 mL), **1c** (0.1 mol L⁻¹), NBu₄NO₃ (0.5 eq.), 30 °C, O₂ atmosphere (20 vol.%), 350 rpm, 4 F (ref. **1c**), 10 mA cm⁻². Sample preparation: three drops of reaction solution were eluted with ethyl acetate through approx. 330 mg silica 60M and filled into a vial for qualitative GC analysis.

Fig. 2: ¹H NMR spectrum (400 MHz, CDCl₃) after electrolysis of cyclooctane (**1c**) to cyclooctanone (**2c**). Electrolysis conditions: as in Fig. 1. Sample preparation: an equimolar amount of internal standard (1,3,5-trimethoxybenzene) regarding **1c** was added and the solvent and starting material was distilled of.

Figure 1 shows that significant amounts of starting material remain unconverted after electrolysis. Nevertheless, not 100% of the mass balance is recovered. One possible explanation is a

fragmentation/decomposition of the starting material or product to unidentified low molecular weight volatile compounds, which appear in the GC in the solvent signal region (within the first 2–2.5 minutes). Looking at Figure 2, it is noticeable that after the reaction, besides the product signals, only solvent, standard, and supporting electrolyte signals are present in significant intensity in the ^1H NMR spectrum (the starting material is distilled off together with the solvent in the rotary evaporator). Accordingly, no conclusions can be drawn about higher molecular weight by-products.

Another possibility for the reduced mass balance could come from the open system resulting from unsealed cells. A permanent gas flow could still carry out portions of starting material, even if oxygen is not bubbled through the solution.

14. Change “The agitation speed of the stirrer within the cell, has a significant influence” to read: “The rate of stirring has a significant influence”

Our response: The sentence has been rephrased: “The stirring rate significantly influences the reaction as a yield drop appears at higher and lower stirring rates than 350 rpm (see Supplementary Table 1).”

15. Be consistent. Sometimes “splitting” is used, other times it’s not. Perhaps “cleavage” would be more appropriate? See “The C=C double bond splitting of cyclic alkenes 5 (Fig. 3) has been discovered by applying the same” and also “mechanistic proposal for alkene double bond splitting”

Our response: Thanks for the comment. The word “splitting” has been exchanged for consistency with “cleavage” throughout the document.

16. Suggest modifying “Following this result, other different membered rings from six to twelve methylene groups were tested in this set-up (Fig. 3a)” to read: “Next, we used this setup to explore the oxidation of cyclic alkanes containing six to twelve methylene groups (Fig. 3a).”

Our response: The sentences have been exchanged accordingly.

17. Change “with” to “in” – “could be synthesised with 19%. As”

Our response: The words has been exchanged accordingly.

18. Use “The resulting superoxide radical anion” rather than: “The hereby formed superoxide radicals,”

Our response: The sentence has been rephrased accordingly.

19. Change “happen” to “occur” in: “aldehydes to carboxylic acids is assumed to happen”

Our response: The words have been exchanged accordingly.

20. Rather than writing “different atmospheres” be specific. Fig. 4b shows “Ar atm”, “air atm”, and “O₂ atm”. Use these labels as “different atmospheres” is unnecessarily vague.

Our response: Thanks for the comment. The authors agree with the reviewer. The vague term “different atmospheres” has been changed to “argon (blue line), air (orange line), and oxygen (grey line) atmosphere” regarding Figure 4i (for Fig. 4 numbering, please consider response 8 to reviewer #2).

21. Change “addressed” to “attributed” in: “This circumstance is addressed to a disturbed O₂ adsorption on the electrode at”

Our response: The words have been exchanged accordingly.

22. Suggest modifying “The experiment has been carried out in a cycling mode, pumping the electrolyte multiple times through the cell into a reservoir, until ...” To read: “These experiments were carried out by recycling the electrolyte through the cell until”

Our response: The sentence has been rephrased accordingly. Please see also response 23.

23. What is meant by “applied charge amount passed through”? Please be more specific.

Our response: This phrase should indicate that the applied charge determines the reaction time here and not the flow rate of the electrolyte through the cell as in single-pass flow electrolysis. The text has been changed for concreteness to: “These experiments were carried out by pumping the electrolyte through the cell in a cyclic manner until a charge amount of 2–4 *F* was applied.”

24. Specify what is meant by “O₂ atmosphere”. It seems to me that there are a number of different scenarios that could be described as being under an O₂ atmosphere.

Our response: The comment is justified because a reaction occurring under ambient air also happens in an O₂ atmosphere. When specifying an O₂ atmosphere, care was taken to ensure that the volume fraction of oxygen in the gas space was also specified (e.g., O₂ atmosphere (100 vol.%)). This information should have been provided in the caption of Figure 4h and has now been added. Thanks for the hint.

25. Modify “The reactions for the substrate variation of the cycloalkenes were carried out in 5 mL PTFE cells under” to read: “The reactions for cycloalkenes 5a-e portrayed in Fig. 3b, were carried out in 5 mL PTFE cells under”

Our response: The sentence has been modified accordingly.

26. Suggest modifying “Besides the synthesis of diacids from disubstituted double bonds, trisubstituted ones lead to formation of α,ω -ketocarboxylic acids, as shown on the example 6e” to read: “In addition to the formation of α,ω -diacids from disubstituted cycloalkenes, α,ω -ketocarboxylic acids 6e can be obtained from trisubstituted alkenes.”

Our response: The sentences have been exchanged accordingly. Instead of “alkenes” at the end, we chose “cycloalkenes” to precisely describe the substrates.

27. Suggest modification of “To investigate mechanistical correlations of the cycloalkane (Fig. 4f) and alkene (Fig. 4h and 4i) oxidation reactions, several experiments have been conducted” to read: “To uncover mechanistic detail, the experiments illustrated in Figures 4f, h, and i were carried out.”

Our response: Figures 4f, h, and i (changed to 4a, b, and c; please consider response 8 to reviewer #2) illustrate schematic reaction pathways but no experiments. Instead, the authors rephrased the sentence: “Instrumental and wet-chemical experiments were carried out to uncover mechanistic detail, which is schematically illustrated in Figures 4a, b, and c.”

28. Modify “atmosphere, whereby among others the nitrated intermediates (10b) and (11b) were detected via GC-MS and illustrated by a mechanistic proposal (Fig. 4k)” to read: “atmosphere, whereby among several unidentified materials, the nitrated intermediates (10b) and (11b) were detected via GC-MS; their formation is illustrated by the mechanistic proposal shown in Fig. 4k.”

Our response: The full sentence has been modified: “The formation of nitrate radicals could be proved via a radical quenching experiment conducted on cyclooctene (5b) under an argon atmosphere. Among several unidentified materials, the nitrated intermediates 10b and 11b were detected via GC-MS; their formation is illustrated by the mechanistic proposal shown in Fig. 4f.” For Figure 4 numbering, please consider response 8 to reviewer #2.

29. Suggest modifying “The C=C double bond splitting of cyclic alkenes 5 (Fig. 3) has been discovered by applying the same methodology” to read: “Application of the same methodology to cyclic alkenes 5 revealed that they undergo C=C double bond cleavage as shown in Fig. 3.”

Our response: The sentences have been exchanged accordingly.

30. Suggest modifying “Following the observation via HPLC-MS of a dicarboxylic acid 6 formation, the reaction conditions were varied using cyclooctene (5b) as model substrate” to read: “Following the observation of a dicarboxylic acid 6 via HPLC-MS, the reaction conditions were varied using cyclooctene (5b) as model substrate”

Our response: The sentences have been exchanged accordingly.

The following is a list of other portions of the text that require work/rewriting/clarification. I leave it to the authors to do so.

Our response: Thanks for the suggestions.

31. Adversely, the method is barely suitable for large technical applications as 1,1,1,3,3,3-hexafluoropropan-2-ol (HFIP) is used as solvent, which global warming potential (GWP₁₀₀) is approximately

Our response: The sentence has been changed: “Adversely, the method is unsuitable for technical scale applications as the solvent, 1,1,1,3,3,3-hexafluoropropan-2-ol (HFIP), has approximately 200 times higher global warming potential (GWP₁₀₀) than carbon dioxide.”

32. Electrolysis was carried out with 4 F regarding 1c and 10 mA/cm² leading to a yield

Our response: The sentence has been changed: “Cyclooctane (1c) was electrolyzed using 10 mA cm⁻² and a charge of 4 F, leading to a 23% yield of 2c.”

33. the ketone yields, compared to the increasing ring sizes, is approximately following the transannular

Our response: The sentence has been changed: “Remarkably, compared to the increasing ring sizes, the general trend regarding the ketone yields is approximately following the transannular Prelog strain, ...”

34. diacid 6c. Despite of that, both substrates can be converted by their own target reactions parallel to each other.

Our response: For better clarification, the sentence has been changed: “Despite that, both substrates can be specifically converted into the ketone or diacid in parallel.”

35. nitrate radicals could be indirectly observed due to a control experiment, conducted

Our response: The sentence has been changed: “The formation of nitrate radicals could be proved via a radical quenching experiment conducted on cyclooctene (5b) under an argon atmosphere.”

36. This observation was supported by anion chromatography, as solely nitrate remains in the aqueous layer subsequently to an extractive work up after the electrolysis (Fig. 4a).

Our response: The sentence has been changed and combined with the one before: “The conclusion that nitrate serves as a mediator due to its reformation after oxidation to a radical is supported by anion chromatography measurements. After electrolysis and an extraction workup, only nitrate

remains in the aqueous layer (Fig. 4g).” For Figure 4 numbering, please consider response 8 to reviewer #2.

37. an increased water content, if the sample contained the hydrocarbon substrate (Fig. 4d).

Our response: The sentence has been changed: “Carrying out electrolysis with cyclooctane (**1c**) substrate revealed an increased water content compared to the one without a substrate after electrolysis (Fig. 4k). Measurements were conducted using a Karl Fischer titration method (see Supplementary Note 3.2.5).” For Figure 4 numbering, please consider response 8 to reviewer #2.

38. For the presented method, a further application field has been demonstrated

Our response: The sentence has been changed: “To demonstrate the broad applicability of the presented method, the substrate scope was successfully extended to toluene substrates **12**.”

39. Reaction monitoring via ¹H NMR spectra recording shows the aldehyde as

Our response: The sentence has been changed: “Reaction monitoring via ¹H NMR spectra shows the aldehyde as an intermediate, which is further converted to the corresponding carboxylic acid when the toluene substrate is almost entirely converted (Fig. 5b).”

40. With lower conditions around 10 mA/cm², 5 F, and 0.02 mol/L of **12**

Our response: The sentence has been changed: “By applying conditions of 10 mA cm⁻², 5 F, and 0.02 mol L⁻¹ of **12**, the reaction can be stopped after the selective formation of the benzaldehyde **13'**. When these parameters are increased to 30 mA cm⁻², 7–12 F, and 0.1 mol L⁻¹ of **12**, the formation of benzoic acid **14** can be promoted.”

41. aldehyde **13'**, while increasing these conditions to 30 mA/cm², 7–12 F and

Our response: The sentence has been changed. Please see response 40.

42. A plausible explanation leads via the formation of acyl radicals (as postulated in Figure 4),

Our response: The sentence has been changed: “Presumably, acyl radicals are formed as described in Fig. 4c, which react with acetonitrile as a radical scavenger and are further oxidized to **14'** (see Supplementary Fig. 19).” For Fig. 4 numbering, please consider response 8 to reviewer #2.

Reviewer #2 (Remarks to the Author):

Waldvogel and co-workers reported the electrochemical valorization of simple alkanes and cycloalkenes into valuable chemicals. The key of the present system is the combined use of NO₃⁻ as a mediator and dioxygen gas, where NO₃⁻ serves as a HAT reagent and O₂ gas enables oxygen functionality. Under the optimized reaction condition, cycloalkanes afforded cyclic ketones, whereas cyclic alkenes afforded dicarboxylated ring-opening products, and product selectivities of these reactions are generally quite high. The authors also demonstrated the successful application of this method to linear fatty acids. NO₃⁻ is used both as a mediator and the electrolyte, and importantly, they demonstrated the recovery of this salt after the reaction, demonstrating the potential high sustainability of the present system.

The present work represents a promising strategy for the sustainable valorization of hydrocarbons, which is still a significant challenge even with state-of-the-art technologies. The yields are not necessarily high (78% in maximum, and mostly ~50%), but that in no way diminishes the value of this

study, given the difficulty of the reactions that challenged the authors. Thus, the reviewer supports the publication of the present study in Nature Communications, after addressing the point listed below.

Our response: The authors gratefully acknowledge the appreciative summary of the paper, the recommendation for Nature Communications and the reviewer's suggestions for improvement.

Scientific points

1. In the abstract, the authors claim “This resource-friendly method is based on nitrate salts in a dual role as anodic mediator and supporting electrolyte, which can be fully recovered and recycled”. However, the manuscript and SI (supporting data 2.7) only demonstrate the recovery, and there is no data of recycle. Is the collected nitrate salt from the reaction mixture still usable?

Our response: The reviewer addresses a valid point. Additional experiments have been performed to cover this claim made in the publication. The results have been implemented in the Supplementary Information as “Supplementary Results 2.7: Recovery and reuse of the supporting electrolyte”. After the extractive workup described in the manuscripts “Method” section, the supporting electrolyte remains in the aqueous phase and can be recovered by evaporation of the water. For the reuse demonstration, two reactions to synthesize dodecanedioic acid (**6c**) from cyclododecene (**5c**) have been performed, while the recovered supporting electrolyte from the first reaction was reused for the second one. Here, a yield drop for **6c** of 17% was observed, which was addressed to possibly remaining water in the supporting electrolyte after extractive workup and drying at 0.04 mbar (70 °C) for 16 h. However, product **6c** was obtained in a yield of 50% with the recycled supporting electrolyte, and ¹H NMR spectra comparison shows no difference in the quality of the supporting electrolyte. The experiments conclude that recovering and recycling NBu₄NO₃ supporting electrolyte for the presented reactions is generally possible.

2. This system relies on the convergent paired electrosynthesis via the oxidation of nitrate anion and the reduction of oxygen gas. The generation of H₂ gas at the cathode as a by-product (especially when using protic solvent) potentially causes an explosion by the co-existence of O₂. This point would be especially crucial when directing large-scale applications for industrial processes. The authors should give some data or discussion in the manuscript.

Our response: The electrochemical generation of hydrogen at the cathode is a frequently encountered counter-reaction to the anodic oxidation of a substrate. An explosive gas mixture of hydrogen and oxygen occurs in a range of approx. 5–95 vol.% H₂ at 1 atm². In our experiments, the steady oxygen flow into the electrolytic cell ensures an oxygen supply for the reaction. It would also carry out hydrogen gas when formed as a by-product or in traces.

To prevent H₂ evolution, we used a glassy carbon cathode, which has a high overvoltage for H₂ evolution^{3,4}. In addition, aprotic polar solvents such as acetonitrile, isobutyronitrile, and dimethyl carbonate are used in the presented method, ensuring diminished protic conditions. However, 2-propanol was used as a co-solvent in flow experiments for diacid synthesis, and water was proposed to emerge during the ketone synthesis.

A Supplementary Note section “3.2.7 Cyclic voltammetry studies” has been added to the Supplementary information to investigate the electrolytes reductive behavior for acetonitrile/water and dimethyl carbonate/2-propanol mixtures in the presence and absence of oxygen. Supplementary Figures 18a and 18b show that oxygen is electrochemically reduced before the solvents in both cases. Therefore, the electrochemical evolution of H₂ from the mentioned protic solvents is considered not to appear. Interestingly, no defined oxygen reduction signal in the DMC/*i*-PrOH system, but a linear current increase has been observed in the applied potential range. It is assumed that the NBu₄⁺ cations

do not act as stabilizing counter ions to the superoxide radicals but that DMC might interact with the reduced oxygen species.

Furthermore, no cathodic gas evolution was observed during the reaction procedure. As mentioned in the manuscript, pH value measurements with indicator paper (Supplementary Figure 16) also show a constant pH before and after electrolysis. Based on these observations and the selected reaction conditions, the authors assume that the evolution of hydrogen gas during the reaction is unlikely.

Regarding the safety aspect of H₂ and O₂, a discussion has been added in the manuscript's Results subsection, "Mechanistic studies." Although the presented method provides good framework conditions for large-scale application in industrial processes, its evaluation is not part of this communication paper. However, it will be considered for future process development.

3. Related to the question above, if there is no competing HER, the reaction media becomes acidic in the late stage of the reaction. The reviewer assumes the acidic condition changes the reactivity of O₂⁻. Are there any data or comments on this point?

Our response: In the case of the oxidation of cycloalkanes to ketones, it is assumed that the intermediate hydroperoxide species eliminate water. The protons released from the oxidative introduction of keto groups into methylene groups thus exit the reaction as water molecules, which accumulate in the reaction solution. A cyclic voltammetry measurement with acetonitrile/water (2 vol.%) with and without the presence of O₂ shows that an O₂ reduction signal appears before the solvent reduction occurs (see Supplementary Fig. 18b). O₂⁻ radical anions can be stabilized by alkylammonium complexes⁵. As O₂⁻ acts as a strong base in aprotic media (pK_a of HO₂ radical ≈ 12 in DMF)⁶, the intermediately formed HNO₃ will be most likely deprotonated, forming nitrate and HO₂ radicals. At this point, the additional formation of H₂O₂ cannot be excluded. In the case of the diacid formation from cyclic alkenes, the reaction media itself stays neutral, as carboxylic acids are formed as products. During the reaction under the given conditions, partial deprotonation of the carboxylic acids might occur via protonation of O₂⁻ radical anions and cannot be excluded.

4. Carboxylic acids generated via the cleavage of alkenes seem to be easily oxidized under electrochemical conditions. Does the oxidation of carboxylic acid compete with the oxidation of nitrate?

Our response: Carboxylic acids are typically treated anodically in the form of their carboxylate anions, as in the example of Kolbe electrolysis. Due to their negative charge, carboxylate anions are more accessible to oxidize than free carboxylic acids. With our method, it is not entirely excluded but rather unlikely that carboxylate anions will be formed. A cyclic voltammetry measurement for the oxidation of adipic acid in the presence of either nitrate or hexafluorophosphate anions has shown that an oxidation signal is just formed for nitrate. In contrast, adipic acid is not oxidized within the applied potential range (see Supplementary Figure 18c). The authors, therefore, assume that the free carboxylic acids formed are not in competition with the electrochemical nitrate oxidation and that the latter is strongly preferred. A corresponding sentence was included in the manuscript's Results subsection, "Mechanistic studies."

5. There is a unique contrast that the olefin cleavage reaction solely afforded dicarboxylic acids, whereas the oxidation of toluenes followed sequential oxidation to benzoic acid via benzaldehyde. Why no aldehyde intermediate was obtained during the olefin cleavage reaction, despite the proposed mechanism in Fig. 4i proceeding via the formation of an aldehyde intermediate?

Our response: The authors thank the reviewer for the observation. Indeed, the olefin cleavage to diacids also leads to aldehydes as intermediates, as shown in the benzylic oxidation. For better

clarification, section 2.9 has been added to the Supplementary information in which an HRMS analysis of the reaction solution for dodecanedioic acid (**6c**) synthesis after electrolysis is shown. Here, the masses of the corresponding aldehyde intermediates and small amounts of chain-shortened diacids were also found. As suggested in Fig. 4c (previously 4i), diacid synthesis also proceeds via aldehyde intermediates.

In contrast to the benzylic oxidation, this work focused on the synthesis of the diacids, which is why the aldehyde species were not isolated. In the case of benzylic oxidation, the focus was on obtaining either the benzaldehydes or the benzoic acids by varying the parameters of the same method, which is why the intermediates were isolated as semicarbazones.

A corresponding note was added to the manuscript "Mechanistic studies" section for the aldehydic intermediates.

6. To pursue large-scale production, the development of a flow system would be the key, and the authors demonstrated promising results (Fig. 3d). The reviewer is wondering how oxygen gas is reacting in the flow cell. As explained in the manuscript (experimental section, GP4), the flow reaction media is segmented. Considering the very small volume of the electrochemical flow cell used herein (electrode gap: 0.050 cm), the introduction of gas seems to be crucial. But in the reality, the reaction proceeds well under the slightly modified conditions from the batch system. Can the authors explain why the segmentation by O₂ gas did not ruin the reaction?

Our response: The reviewer raises a good point regarding process engineering considerations. The process at hand requires a sufficient concentration of dissolved oxygen. According to the authors, the reaction works with a segmented flow under the given conditions mainly due to the following three points.

- 1) The segmented flow can promote the mass transfer of a gas-liquid system as reported in the literature^{7,8}: quick liquid and gas flow rates in small channels promote the internal circulation within liquid slugs, leading to an enhanced mixing within the liquid phase and an increased mass transfer. Moreover, within the applied flow rates, the residence time of the gas-liquid system in the electrochemical cell is short at approx. 1–2 seconds (calculated from the cell volume of 0.6 mL and a flow rate of 20 mL/min).

- 2) The electrolyte, stored in a reservoir, was pumped through the electrochemical cell multiple times. When rinsing back into the reservoir, the gas phase exiting the tubing was directed into the electrolyte so that gas bubbles were sparged through, ensuring an increased gas-liquid interfacial area.

- 3) The electrolyte inlet and outlet design of the flow cell allows distribution/spreading and possibly turbulences of the electrolyte stream through fine channels as it passes through the cell (see Fig. 3), which might influence and increase the gas-liquid interfacial area.

Fig. 3: Schematic illustrations of a PTFE half-cell block of the flow cell used. **a** Cross-section through fine inlet/outlet channels for electrolyte distribution. **b** Cross-section through the half-cell block / side view of the electrolyte inlet/outlet channels.

7. Is there any product formed in the absence of O₂ (Fig. 2, Entry 3)? The reviewer is curious if a desaturation reaction such as olefin formation occurs if there is no O₂.

Our response: For illustration, gas chromatographic analyses are shown in Fig. 4., below. As can be seen, mainly starting material could be detected after electrolysis. No significant conversion of the starting material was observed in the absence of oxygen. If small amounts of oxygen are present in the gas space in the cell, small amounts of ketone and alcohol of the respective cycloalkane compound can be formed. In the case of cycloalkenes, small amounts of oxygen may lead to traces of the corresponding epoxide compound, as was also observed in the control experiment under an argon atmosphere (see manuscript, Fig. 4f).

In contrast, desaturation of the cycloalkanes to cycloalkenes/olefins was not observed. During electrolysis without oxygen, an electrolyte's rapid color change occurs from colorless to dark yellowish/brownish. It is therefore assumed that the electrolyte decomposes during the process. At the same time, a significant voltage increase is observed when oxygen is not present in the system, which underlines the electroactive role of O₂ (from about 3.5 V (O₂ 100 vol.%) to about 5.7 V (O₂ 0 vol.%) initial voltage, electrolyte: acetonitrile, NBu₄NO₃ (0.1 M), glassy carbon electrodes, 0.5 cm electrode gap, 10 mA cm⁻²).

Fig. 4: **a** Electrolysis of cyclooctane (**1c**) under an inert gas atmosphere (N₂). Traces of cyclooctanol (**3c**) were observed. **b** Electrolysis of cyclooctene (**5b**) under an inert gas atmosphere (N₂). As an impurity

in the starting material, cyclooctane (**1c**) is present. Small amounts of epoxide (**41**) and cyclooctanone (**2c**) were observed via GC-MS, probably caused by remaining amounts of oxygen in the system.

Non-scientific points

8. Figures in the manuscript should appear following alphabetical order. For example, Fig. 4k appears first in the manuscript before mentioning Figs. 4a-j. It makes the manuscript hard to read. Context and Figures should be organized.

Our response: The authors agree that the appearance of figures in the text should follow the alphabetical order of the figures themselves. The following changes have been made:

1. The text passage describing Fig. 3e has been placed below the passages of Figures 3c and 3d.
2. The alphabetical numbering of the individual Figures in Figure 4 were changed as follows, and Figure 4, including the Figure description, was adjusted accordingly.

4f → 4a

4h → 4b

4i → 4c

4e → 4d

4j → 4e

4k → 4f

4a → 4g

4g → 4h

4b → 4i

4c → 4j

4d → 4k

9. Line 32, a comma is missing after "...application fields...".

Our response: Due to rephrasing parts of the manuscript text, line 32 shifted to line 25. The missing comma has been inserted.

10. Line 83 should be "oxo-functionalization", not "oxo-functionalisation"

Our response: The spelling has been changed from British English to American English throughout the manuscripts. The word "oxo-functionalization" has been corrected.

11. Line 114, a comma is missing after "cell...".

Our response: The corresponding sentence has been rephrased due to linguistic improvements: "Furthermore, polar solvents with high permittivity are required to avoid high ohmic resistance within the electrochemical cell."

12. Line 115, "showing" should be "show".

Our response: The respective sentence has been rephrased. Please see response 10 to reviewer #1.

13. In line 119 (... Inspired by literature, ...), it is unclear which literature the authors indicate.

Our response: Thanks for the hint. The respective literature has been referenced. Due to rephrasing parts of the manuscript text, line 119 shifted to line 105.

Reviewer #3 (Remarks to the Author):

Authors describe the use of nitrate salts with a dual role as catalyst and electrolyte of these reactions. The oxidation of alkanes is a challenging transformation due to the lack of activating groups in these molecules. On the other hand, the selective oxidative cleavage of alkenes provides high yields of the corresponding diacids.

Our response: The authors agree with the reviewer's comment. The conversion of alkanes is challenging, and the results, especially for the diacid synthesis from cycloalkenes, are promising.

Apart from this comments, I have a few more curiosities:

1. I miss a couple of examples of branched alkanes in the first part. Would the tertiary carbons get oxidized easier and selectively? Would the alcohols be obtained as sole products? Trying this reaction with a couple of more complicated alkanes would be illustrative. The same in the oxidative cleavage of alkenes. Would a more substituted cyclic alkene give the corresponding hydroxy acid?

Our response: The authors welcome the idea of testing the method on branched substrates. The following additions have been made to the publication to address the reviewer's comment:

To illustrate the selectivity of the reaction towards branched alkanes, an additional section, "2.8 Oxo-functionalization results with branched cycloalkanes," was added to the Supplementary Information. Here, the selectivities concerning the formation of alcohols and ketones and their regioselectivities for the oxidized molecular sites on methylcyclohexane (**16**), ethylcyclohexane (**24**), and *trans*-decalin (**31**) are shown. The selectivity ratios were determined by GC-FID integrals and product assignments by GC-MS signal comparison with NIST17 mass spectral library entries. As can be seen from this section, the selectivity of the presented reaction generally goes more toward the formation of ketones than alcohols. For methylcyclohexane (**16**), the tertiary alcohol **17** is formed with a selectivity comparable to 3-methylcyclohexanone (**22**). Interestingly, for ethylcyclohexane (**24**), the selectivity towards the tertiary alcohol decreases, probably due to the longer chain's steric effects. In principle, the reaction conditions for branched alkanes lead to several oxidation products. A corresponding sentence has been added to the manuscript.

In the case of cycloalkenes, an example of a branched natural product was tested using *S*-(-)-limonene, resulting in an isolated yield of the corresponding ketocarboxylic acid **6f** of 17%. This value is comparable to adipic acid (**6a**, 19%) from cyclohexene (**5a**). No corresponding hydroxy acid at the branching position was observed. The example has been added to Fig. 3 in the manuscript, and its characterization has been added to the Supplementary Information. A corresponding sentence has been added to the manuscript.

2. In the mechanism of the oxidation of toluene derivatives, aldehyde is formed as intermediate, which afterwards gives the acid. Is this not observed in the case of the alkene oxidative cleavage?

Our response: The authors thank the reviewer for the pointer. Indeed, aldehydes are also formed as intermediates in the diacid synthesis from cycloalkenes. For clarification, section 2.9 was included in the Supplementary information, where corresponding aldehydes were detected via HRMS analysis as intermediates after electrolysis of cyclododecene (**5c**). Please also consider response 5 to reviewer #2.

3. The authors say that 'Inspired by literature' (line 119) but no reference is mentioned here.

Our response: Thanks for the hint. The respective literature has been referenced. Due to rephrasing parts of the manuscript text, line 119 shifted to line 105.

4. Why is a lower temperature (5°C) beneficial for the reaction?

Our response: Observations showed that, in general, both reactions, ketone synthesis, and diacid synthesis, proceed best at temperatures in the range 20–30 °C. Fig. 2, entry 8 in the manuscript shows one of the highest achieved yields of octanedioic acid (**6b**) with 47% at 5 °C. Nevertheless, the yields are comparable with lower substrate concentrations at 30–35 °C (see Fig. 2, entries 11 and 13). Furthermore, lower yields for 5 °C were obtained in optimization experiments for dodecanedioic acid (**6c**) syntheses (see Supplementary Results 2.3). In principle, lower reaction temperatures improve oxygen solubility in the solvent. At the same time, the risk of substrate evaporation is reduced, which could explain the good yields in the case of **6b**. In the case of converting norbornene to cyclopentane-1,3-dicarboxylic acid (**6d**), a temperature of 5 °C improved the selectivity of the reaction. When the reaction was carried out in the range of 20–30 °C, the product was observed only in traces via HPLC-MS. The authors suggest that in this particular case, the high ring strain of norbornene (about 20 kcal/mol, comparable to cyclobutane compounds)⁹ possibly affects the conversion. The release of ring strain energy can lead to an exothermic reaction that is better mediated at lower temperatures. In addition, cooling the reaction solution prevents the evaporation of the relatively volatile starting material.

5. Authors tried using BDD and graphite electrodes as well, but no combination of them was tested. Sometimes, a different material in anode and cathode can positively affect to the reaction performance.

Our response: The authors agree with the reviewer. Sometimes different electrode materials as anode and cathode can lead to the preference of a specific reaction. Therefore, additional experiments were performed according to the reviewer's point, listed in the table below.

Table 1: Additional experiments using combinations of electrodes, according to GP1.

Entry	Deviation from conditions ^a	2c ^b	3c ^c	4c ^c
1	None	31%	1%	6%
1	GC BDD	17%	0%	1%
2	BDD GC	10%	1%	1%
3	GC graphite	25%	1%	3%
4	BDD graphite	12%	1%	1%
5	graphite GC	8%	2%	1%

^aUndivided 5 mL PTFE cell, glassy carbon electrodes, acetonitrile (5 mL), **1c** (0.2 mol L⁻¹), NBu₄NO₃ (0.5 eq.), 30 °C, O₂ atm. (20 vol.%), 350 rpm, 4 F, 10 mA cm⁻². ^bYield determination via GC (external calibration of **2c**, 1,3,5-trimethoxybenzene as internal standard). ^cYield determination via GC integrals calculated based on yield of **2c**.

The combination of GC || graphite (**2c**, 25%) gave the best result in the additional tests. Generally, the results are in the same yield range as when BBD || BDD (**2c**, 20%) or graphite || graphite (**2c**, 15%) are used, probably due to the common carbon-based nature of the materials. Using graphite as an anode material is generally not recommended due to its low stability in this reaction, as it decomposes promptly by detaching black particles.

6. For the oxidation of alkanes, the optimized conditions were found using 20 vol% oxygen. However, the reaction is performed with pure oxygen atmosphere, and this could be due to 'larger contact area between the oxygen containing atmosphere and the reaction solution. Was it re-tested using air or 20 vol% in O₂?

Our response: Indeed, the reaction of cyclooctane (**1c**) as model substrate was also carried out with air in a round bottom flask setup applying the same electrochemical parameters. However, the yield of cyclooctanone (**2c**) was 35%. After using an increased oxygen concentration, the yield could be improved, as described in the manuscript. This observation was added as additional information to the manuscript's text.

7. Figure 4: this reviewer suggests a better organization of the panels of this figure. The text and the figure have a different organization and, since the manuscript deals with two reactions, reading this mechanistic part with the panels unorganized regarding the text, is a bit confusing.

Our response: The authors agree that Figure 4 needs to be organized better. The numbering, as well as the alignment, has been adjusted in alphabetical order according to the mention in the text. Please see response 8 to reviewer #2.

8. In the case of running the reaction under pure N₂, no desired products are observed. Do the starting materials remain untouched?

Our response: In the case of electrolysis under inert gas conditions, such as under N₂ atmosphere, the starting materials remain largely unreacted, according to our observations. Instead, it is probably primarily the electrolyte that is decomposed into unidentified compounds. In the presence of remaining small amounts of oxygen, traces of oxidation products are detected. For this, please also see answer 7 to reviewer #2.

9. Authors mention (line 250) that 'in the case of cycloalkanes, the nitrate radicals serve as mediator, as the anion is reformed'. 0.5 eq of this nitrate salt are used, but yields are always below 50%, which means that the TON of this mediator is lower than 1. Is starting material remained untouched in this reaction? In this regard, authors mention that 'The reaction proved itself to be outstandingly selective, as only cyclooctanol (**3c**) and cyclooctane-1,4-dione (**4c**) were obtained as significant by-products with yields of 2% and 3%.' (lines 128-129). However, only 23% yield is obtained. What happens with the rest of material? If no starting material is recovered, I would not affirm that the method has an 'outstanding' selectivity of the method.

Our response: The reviewer stresses a valid point. After the reaction, unconverted amounts of starting material remain. In the case of cyclooctane (**1c**) at 20 vol.% O₂ approx. 10–30% and at 100 vol.% O₂ approx. 30–50%. With regards to the yields of cyclooctanone (**2c**), the mediator's TON generally lies in a range of 0.14–0.37 (100 vol.% O₂) and 0.11–0.62 (20 vol.% O₂), respectively. However, TONs of 1.28 and 1.15 have also been observed, using the mediator in an equivalent of 0.2 instead of 0.5 under 20 vol.% O₂.

Nevertheless, a reduced mass balance remains after the reaction. Possible explanations for this are, for example, the formation of unidentified volatile fragmentation/decomposition products or the increased expulsion of the starting materials by the gas flow. In this regard, please also consider our

response 13 to reviewer #1. The emphasis on outstanding selectivity has been relativized by crossing out the term "outstanding" even though the number of significant detectable by-products is reasonably limited.

10. Authors claim that the electrolyte can be recovered and recycled, but no recycling was tested.

Our response: The authors thank the reviewer for the comment. Following response 1 from reviewer #2, the recyclability of the supporting electrolyte was addressed and added to the Supplementary Information (see Supplementary Result 2.7). The recovery and recycling of the supporting electrolyte was demonstrated in two experiments for the reaction of cyclododecene (**5c**) to dodecanedioic acid (**6c**). After reusing the recovered salt, **6c** was obtained with a yield of 50%. The additional experiments showed that general recovery and recycling of NBu_4NO_3 supporting electrolyte for the presented reactions is feasible.

11. In the obtention of carboxylic acid, authors observe dehomologation only in the case of **6e**. Was it not observed in any other case? Dehomologation of acids is common under electrochemical conditions due to overoxidation.

Our response: The reviewer raises a good point. Generally, dehomologation cannot be entirely excluded for other diacids. In the case of **6e**, dehomologation product was identified at least in significant amounts. Nonetheless, also in the case of dodecanedioic acid (**6c**) synthesis chain-shortened products were observed by HRMS analysis after the reaction (see Supplementary Figure 13). However, the amount of dehomologation products appear to form only to a very small extent in this reaction. The diacid products of the unsubstituted cycloalkenes could be obtained extractive without column chromatographic purification, and no chain-shortened acids were observed in the product.

12. Supplementary Figure 8: indicate the space of time between each NMR shown. Indicate that the time increases going up in the NMRs.

Our response: Thanks for the comment. The time indication in the respective Supplementary Figure 8 has been added.

REFERENCES

1. Criegee, R. Mechanism of ozonolysis. *Angew. Chem. Int. Ed.* **14**, 745–752 (1975).
2. Frawley, T. J. Safety in a Green Hydrogen Economy. *Fauske & Associates, Inc.* <https://www.fauske.com/blog/safety-in-a-green-hydrogen-economy> (2021).
3. Heard, D. M. & Lennox, A. J. J. Electrode materials in modern organic electrochemistry. *Angew. Chem. Int. Ed.* **132**, 19026–19044 (2020).
4. Hasan, M. M. & Allam, N. K. An alternative, low-dissolution counter electrode to prevent deceptive enhancement of HER overpotential. *Sci. Rep.* **12**, 9368 (2022).
5. Zimmermann, M. Oxygen reduction reaction mechanism on glassy carbon in aprotic organic solvents. (Université Grenoble Alpes, Grenoble, 2015).
6. Sawyer, D. T. & Valentine, J. S. How super is superoxide? *Acc. Chem. Res.* **14**, 393–400 (1981).
7. Tan, J., Lu, Y. C., Xu, J. H. & Luo, G. S. Mass transfer performance of gas–liquid segmented flow in microchannels. *Chem. Eng. J.* **181–182**, 229–235 (2012).
8. Noël, T., Cao, Y. & Laudadio, G. The fundamentals behind the use of flow reactors in electrochemistry. *Acc. Chem. Res.* **52**, 2858–2869 (2019).
9. Khoury, P. R., Goddard, J. D. & Tam, W. Ring strain energies: substituted rings, norbornanes, norbornenes and norbornadienes. *Tetrahedron* **60**, 8103–8112 (2004).

REVIEWERS' COMMENTS

Reviewer #1 (Remarks to the Author):

The authors have carried out a thorough and detailed assessment of the comments and concerns of each reviewer. They have satisfactorily addressed each of my concerns, and I believe they have done so for the other reviewers as well. I recommend publication of this important piece of research.

Reviewer #2 (Remarks to the Author):

The authors addressed all the questions, and the reviewer is satisfied with their response. The quality of the manuscript has been improved, and thus the reviewer recommends the acceptance of the manuscript in its current form.

Reviewer #3 (Remarks to the Author):

All concerns have been solved. I recommend this revised manuscript for publications

REVIEWERS' COMMENTS

Reviewer #1 (Remarks to the Author):

The authors have carried out a thorough and detailed assessment of the comments and concerns of each reviewer. They have satisfactorily addressed each of my concerns, and I believe they have done so for the other reviewers as well. I recommend publication of this important piece of research.

Our response: The authors are pleased that the reviewer is satisfied and thank the reviewer for recommending the manuscript for publication.

Reviewer #2 (Remarks to the Author):

The authors addressed all the questions, and the reviewer is satisfied with their response. The quality of the manuscript has been improved, and thus the reviewer recommends the acceptance of the manuscript in its current form.

Our response: The authors are pleased that the reviewer is satisfied and thank the reviewer for recommending the manuscript for publication.

Reviewer #3 (Remarks to the Author):

All concerns have been solved. I recommend this revised manuscript for publications

Our response: The authors are pleased that the reviewer is satisfied and thank the reviewer for recommending the manuscript for publication.